# Hierarchical mechanism of amino acid sensing by the T-box riboswitch

Krishna C. Suddala[1,2], Javier Cabello-Villegas[2], Malgorzata Michnicka[3], Collin Marshall[2], Edward P. Nikonowicz[3] & Nils G. Walter [2]

In Gram-positive bacteria, T-box riboswitches control gene expression to maintain the cellular pools of aminoacylated tRNAs essential for protein biosynthesis. Co-transcriptional binding of an uncharged tRNA to the riboswitch stabilizes an antiterminator, allowing transcription read-through, whereas an aminoacylated tRNA does not. Recent structural studies have resolved two contact points between tRNA and Stem-I in the 5′ half of the T-box riboswitch, but little is known about the mechanism empowering transcriptional control by a small, distal aminoacyl modification. Using single-molecule fluorescence microscopy, we have probed the kinetic and structural underpinnings of tRNA binding to a glycyl T-box riboswitch. We observe a two-step mechanism where fast, dynamic recruitment of tRNA by Stem-I is followed by ultra-stable anchoring by the downstream antiterminator, but only without aminoacylation. Our results support a hierarchical sensing mechanism wherein dynamic global binding of the tRNA body is followed by localized readout of its aminoacylation status by snap-lock-based trapping.

[1] Biophysics, University of Michigan, Ann Arbor, MI 48109, USA. [2] Single Molecule Analysis Group, Department of Chemistry, University of Michigan, Ann Arbor, MI 48109, USA. [3] Department of Biochemistry and Cell Biology, Rice University, Houston, TX 77005, USA. Correspondence and requests for materials should be addressed to E.P.N. (email: edn@rice.edu) or to N.G.W. (email: nwalter@umich.edu)

Riboswitches are structured domains present in the 5′-untranslated region (UTR) of certain bacterial messenger RNAs (mRNAs) that regulate the expression of downstream genes by directly sensing their metabolic products[1,2]. Almost 40 different classes of riboswitches have been discovered that recognize diverse ligands such as amino acids, nucleobases, cofactors, second messengers, metal ions, transfer RNAs (tRNAs), and can also sense pH and temperature changes[1]. Gene expression is primarily regulated either through premature transcription termination or through inhibition of translation initiation via sequestration of the ribosome binding site[2]. Gram-positive bacteria, including pathogenic *Bacillus* strains, often control the expression of genes involved in maintaining aminoacyl-tRNA pools—aminoacyl-tRNA synthetases, amino acid biosynthesis enzymes, and membrane transporters—through a structure located in the 5′-UTR, or leader, of mRNA termed a T-box riboswitch[3–5]. Binding of a non-aminoacylated (uncharged) transfer RNA (tRNA) to the T-box in *trans* is thought to stabilize an antiterminator, promoting transcription read-through by preventing formation of the competing terminator structure[3–5] (Fig. 1a). Although the T-box RNA was the first riboswitch to be discovered[6] and shares features with many known and emerging bacterial mRNAs controlled by regulatory RNAs in *trans*[7–9], its large size and that of its tRNA ligand have impeded biophysical studies, making it mechanistically relatively unexplored. Recent

success with small-molecule drugs against the class of flavin mononucleotide (FMN) riboswitches suggests that additional antibiotic therapies could arise if the structure–dynamics–function relationships of a greater diversity of riboswitches, including T-boxes, were understood[10–12].

The archetypal *glyQS* T-box riboswitch from *Bacillus subtilis* controls the expression of *glyQS* gene encoding glycyl-tRNA synthetase in response to the ratio of charged to uncharged tRNA[Gly] in the cell[13]. The *glyQS* riboswitch is composed of Stem-I and Stem-III and the mutually exclusive antiterminator and terminator structures (Fig. 1a, b)[13]. Recent structural studies of Stem-I:tRNA complexes have revealed two conserved features of cognate tRNA selection: a sequence-specific interaction between the Specifier sequence and the tRNA anticodon, and a stacking interaction between a double T-loop motif (DTM) and the tRNA elbow that effectively acts as an architectural caliper[14–17] of the long side of the tRNA's "L" shape (Fig. 1c). A third interaction between a highly conserved 7-nucleotide bulge of the antiterminator (the T-box sequence) and the 3′ acceptor end of tRNA stabilizes the antiterminator. This later interaction precludes formation of the thermodynamically more favorable terminator by sequestering nucleotides required for its formation (Fig. 1a, b). Stable formation of this interaction is thought to be dependent on the tRNA's aminoacylation state (Fig. 1a); however, details of the amino acid sensing mechanism are still lacking. Stem-I in the

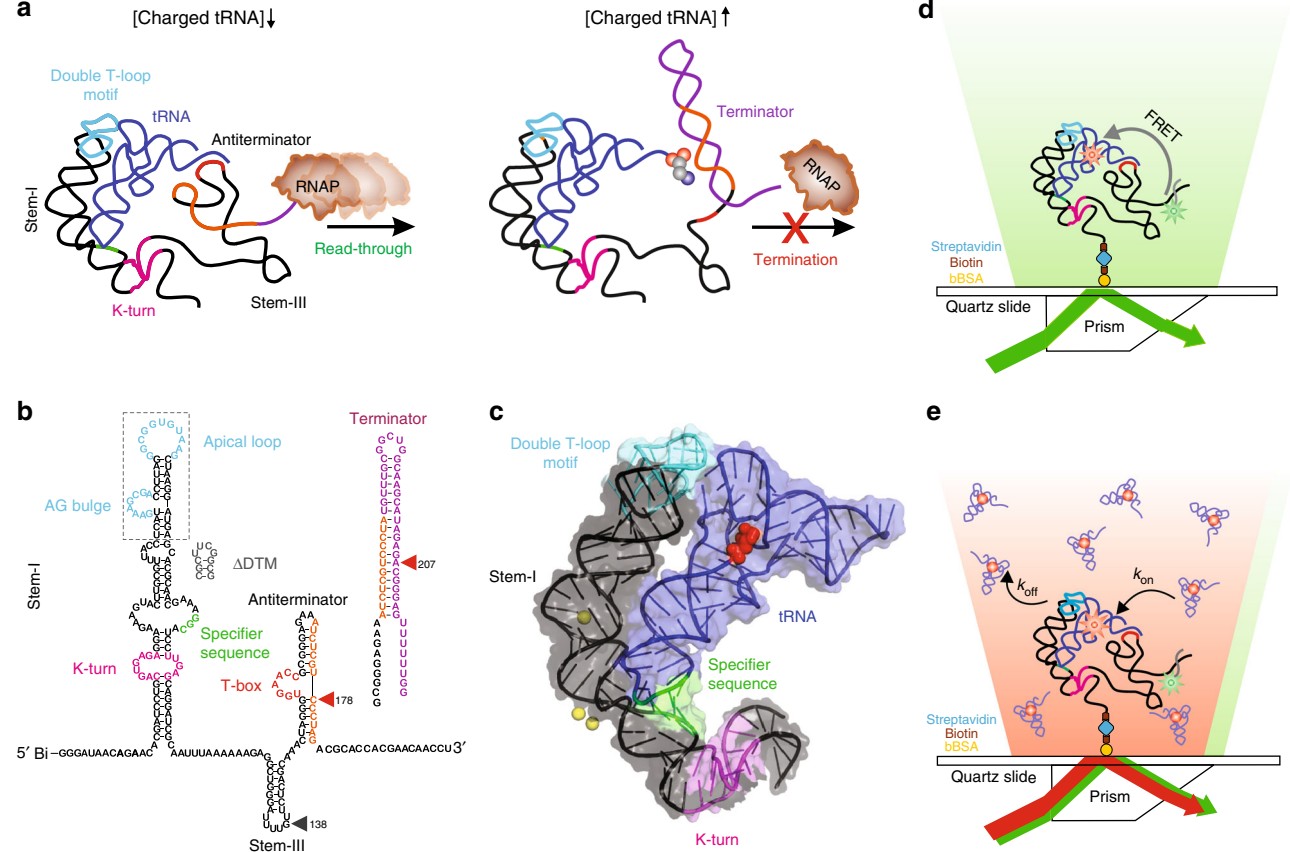

**Fig. 1** Single-molecule analysis of the *B. subtilis glyQS* T-box riboswitch. **a** General mechanism of transcriptional regulation by the T-box riboswitch. **b** Sequence and secondary structure of the *glyQS* T-box (core) riboswitch construct used in this study with key regions highlighted in color. The ΔDTM construct lacks the distal sequences in the boxed region and is capped by a UUCG tetraloop sequence (black). The sequence of the mutually exclusive terminator hairpin is also shown. The positions of the major in vitro and in vivo pause sites observed are marked by gray and red arrows, respectively. **c** Ribbon and surface representation of an *Oceanobacillus iheyensis*-derived Stem-I:tRNA crystal structure (PDB ID: 4LCK). Red spheres depict the Cy5 label at position 46 in the tRNA variable loop. $Mg^{2+}$ ions bound near the dinucleotide bulge and the Specifier sequence are depicted as yellow spheres. **d** Schematic of the prism-TIRF-based single-molecule FRET setup showing an immobilized T-box:tRNA complex. **e** Schematic of the single-molecule fluorescence experiment to measure the binding kinetics in the presence of free Cy5-tRNA[Gly]

5′-half of the T-box riboswitch has been suggested to play an early role in the regulatory mechanism by recruiting the tRNA via its two contact points[18]. Once transcribed, the antiterminator would then need to rapidly interrogate the tRNA aminoacylation status to exert kinetic control over the transcription outcome, as has been suggested for riboswitches with smaller ligands[19,20]. However, the underlying kinetics of tRNA binding to the T-box riboswitch and the relative contribution of each of the three recognition elements remain unknown. Furthermore, the mechanism of how aminoacylation of tRNA prevents transcription read-through (i.e., antitermination) is unclear.

Here, we combine single-molecule fluorescence energy transfer (smFRET)[21,22] probing of the dynamic solution conformations of the natively folded glyQS T-box riboswitch core (including Stem-I, Stem-III, and the antiterminator, Fig. 1b), alone and in complex with tRNA (Fig. 1d) with single-molecule fluorescence assays that kinetically characterize the interaction of the two RNA species (Fig. 1e). Our aim is to delineate the contributions of the individual T-box core:tRNA contact points, the tRNA$^{Gly}$ aminoacylation state, and the complexation of EF-Tu with aminoacylated tRNA$^{Gly}$ to ligand sensing kinetics. Our results show that the T-box riboswitch core adopts a pre-organized, hinged, architecturally stable conformation that is not significantly altered by tRNA binding. Stem-I suffices to recruit the tRNA with a fast association rate constant ($k_{on}$), yet also exhibits a fast dissociation rate constant $k_{off}$. The T-box:tRNA complex becomes highly stable (in the following referred to as "ultra-stable") only after the antiterminator sequence is available for tRNA binding. Our kinetic measurements revealed three distinct $k_{off}$ values. Using core length variants, we show that $k_{off}^{fast}$, $k_{off}^{medium}$, and $k_{off}^{slow}$ correspond to tRNA primarily engaging with the Specifier and DTM motifs, the Specifier and antiterminator motifs, and all three motifs, respectively. We found that the DTM accelerates tRNA binding, a rate that needs to be fine-tuned in each species for effective co-transcriptional regulation, while this contact is also transient and displays molecule-to-molecule heterogeneity. Strikingly, the small glycyl modification only slightly accelerates $k_{off}^{medium}$ but leads to a dramatic loss of the ultra-stable complex characterized by $k_{off}^{slow}$. Similarly, formation of the ternary aa-tRNA:EF-Tu:GTP complex specifically suppresses this ultra-stable tRNA anchoring on the T-box, demonstrating that T-box riboswitches have evolved to sense even the smallest glycyl modification of a tRNA independent of its association with EF-Tu. We propose a hierarchical sensing model wherein rapid binding of the tRNA body by Stem-I is followed by co-transcriptional readout by the emerging antiterminator sequence, aided by a strong transcriptional pause. This T-box sequence is pre-positioned by a designer hinge to dynamically sense the absence of tRNA aminoacylation through a structural snap-lock mechanism.

## Results

### Stem-I recruits tRNA fast via both the Specifier and DTM.

To first probe the solution conformation of the previously crystallized Stem-I:tRNA complex, we performed smFRET (Fig. 1d) between a tRNA$^{Gly}$ labeled with Cy5 at U46 in the variable loop and a 5′-Cy3-labeled locked nucleic acid (LNA) oligonucleotide stably hybridized near the base of the natively purified, 5′-biotinylated Stem-I fragment of the glyQS T-box riboswitch (Supplementary Fig. 1). We did not detect surface-bound Cy5-tRNA$^{Gly}$ after incubation of the dilute stoichiometric reaction mixture in standard buffer (10 mM Tris-HCl, pH 6.1, 50 mM KCl, 10 mM MgCl$_2$) with a streptavidin-coated slide surface, suggesting complexes of only short lifetime. Therefore, we added excess (25–50 nM) Cy5-tRNA$^{Gly}$ into the microfluidic channel to detect any transient interactions to the immobilized Cy3-LNA-labeled

Stem-I. Single-molecule traces indeed displayed repeated short (<10 s) binding events with a relative FRET efficiency ($E_{FRET}$) of ~0.3 (Fig. 2a). The ensemble FRET histogram displayed a single peak with a mean of 0.28 ± 0.13 (Fig. 2b), corresponding to a mean distance of ~63 Å between the fluorophores (based on a Förster radius of 54 Å [23]), a distance consistent with the available Stem-I:tRNA co-crystal structures[15,16]. In a few traces, brief transitions to high FRET ($E_{FRET}$ ~0.85; corresponding to an ~40 Å distance) were observed before the tRNA dissociated (Supplementary Fig. 2b), suggesting that the tRNA:Stem-I complex can perform an occasional closing motion, presumably requiring a transient loss of the elbow interaction with the DTM, which then leads to tRNA dissociation.

To measure the tRNA binding kinetics and detect all binding events, including those involving any zero or low FRET states, we switched to single-molecule fluorescence assays colocalizing tRNA and T-box molecules within diffraction limited spots (Fig. 1e), conceptually similar to our single-molecule kinetic analysis of RNA transient structure (SiM-KARTS)[24]. In these experiments, Cy5 on the tRNA was directly excited to detect binding events to the Cy3-detected T-box, manifesting as sharp spikes in intensity over background that enable the measurement of individual bound ($t_{bound}$) and unbound ($t_{unbound}$) dwell times (Fig. 2c). Single-molecule traces revealed that tRNA binds transiently to Stem-I with short dwell times, $t_{bound}$ (Fig. 2d), similar to our observations from smFRET. The cumulative distributions of dwell times ($t_{unbound}$) and ($t_{bound}$) fit well with single-exponential functions (Fig. 2e, f), yielding a binding rate constant $k_{on} = 0.79 \pm 0.02 \times 10^6$ M$^{-1}$ s$^{-1}$ (lifetime $\tau_{unbound} = 49.7 \pm 3.7$ s at a concentration of 25 nM tRNA) and a dissociation rate constant $k_{off} = 0.24 \pm 0.02$ s$^{-1}$ ($\tau_{bound} = 4.5 \pm 0.2$ s), respectively, defining a $K_d$ of ~300 nM. This $K_d$ is within ~2–3-fold of the values previously measured with isothermal titration calorimetry for the B. subtilis and Oceanobacillus iheyensis glyQS Stem-I under different buffer conditions[15,25]. The fast $k_{on}$ and $k_{off}$ allowed kinetics to be measured using FRET within the time window before Cy3 photobleached, yielding consistent and homogeneous kinetics (Supplementary Fig. 2). Similarly, we verified that the $\tau_{bound}$ of ~4.5 s was significantly shorter than the average Cy5 photobleaching lifetime of ~64 s under our imaging conditions, and that non-specific binding of tRNA to the passivated slide surface was negligible (Supplementary Fig. 3). As a control, a Stem-I variant with an 18-bp stem inserted between the apical loop and AG bulge to prevent DTM formation did not exhibit any tRNA binding events (Supplementary Fig. 2). Taken together, these data indicate that Stem-I recruits tRNA efficiently with a fast $k_{on}$, but suffers from a fast $k_{off}$; a feature masked in previous equilibrium affinity measurements and crystallization[15,16].

### glyQS T-box riboswitch adopts a pre-organized conformation.

To probe the solution conformation of the T-box core riboswitch, we hybridized Cy5-labeled LNA and Cy3-labeled DNA oligonucleotides to the 5′-end and 3′-end, respectively (Fig. 3a). The resulting FRET histogram in standard buffer displayed one major and two minor peaks with $E_{FRET}$ values of 0.52 ± 0.12 (90%), 0.09 ± 0.12 (9%), and 0.92 ± 0.07 (1%), corresponding to Stem-I-to-antiterminator distance estimates of ~53, ~79, and ~36 Å (each ±5 Å), respectively (Fig. 3a). Most traces showed either a stable 0.52 or a stable 0.09 FRET state, with a small (~8%) fraction displaying brief 0.52-to-0.92 FRET transitions (Fig. 3b). Addition of a saturating concentration (2 μM) of unlabeled tRNA$^{Gly}$ ($K_d$ ~50 nM, this study and ref. [15]) led to no significant increase in the mean value of the major mid-FRET state (0.57 ± 0.12) or change of the FRET distribution (Fig. 3a). tRNA addition did, however,

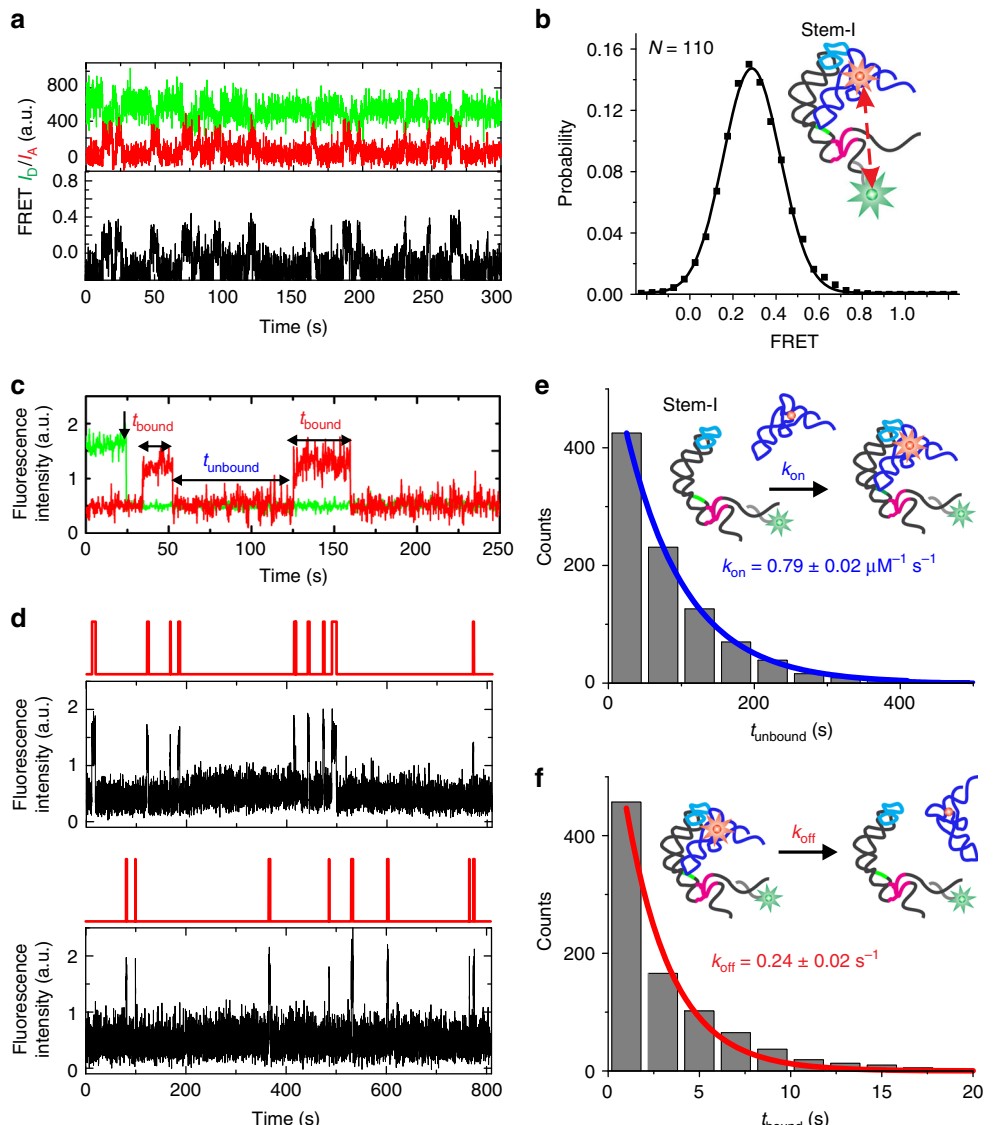

**Fig. 2** smFRET and kinetic analysis of tRNA binding to Stem-I. **a** Example smFRET trace of Cy5-labeled tRNA$^{Gly}$ binding to Stem-I hybridized with Cy3-labeled DNA oligonucleotide at the 3′-end. Green, Cy3 intensity; red, Cy5 intensity; black, FRET efficiency. **b** Population FRET histogram generated from the binding events and fit with a single Gaussian function (black). The mean FRET value given by the center of the peak provides an estimate of the distance between the fluorophores, which in this example is between the base of Stem-I and U46 of the bound tRNA. **c** Example binding trace showing Cy3 intensity that photobleaches in a single-step indicative of one T-box riboswitch molecule, and abrupt increases in Cy5 intensity, indicative of tRNA binding and dissociation events from which dwell times in bound and unbound states, $t_{bound}$ and $t_{unbound}$, can be obtained. **d** Representative single-molecule traces showing tRNA binding to Stem-I. The Cy5 signal is shown in black and a two-state hidden Markov model (HMM) idealized trace is shown on top in red. Dwell-time histograms for **e** $t_{unbound}$ and **f** $t_{bound}$ were fit with single-exponential functions (blue and red), to estimate the rate constants, $k_{on}$ and $k_{off}$, respectively. Errors are s.d. of at least three independent replicates

result in a loss of both the small fraction of dynamic traces and the high FRET population, as clearly evident in transition density plots (Fig. 3b, Supplementary Fig. 4). These data show that the T-box structure is largely pre-organized along its antiterminator-to-Stem-I vector prior to tRNA binding.

Next, we interrogated the distance between the base of Stem-I and the tRNA variable loop within the context of the T-box core (Fig. 3c). The FRET histogram for this complex revealed a dominant (61%) state with $E_{FRET}= 0.38 \pm 0.12$ (~59 Å; consistent with the measurement for the isolated Stem-I, Fig. 2b, and in reasonable agreement with the ~64 Å observed in the crystal structures[13,14]) and a relatively minor (39%) state with $E_{FRET}$ ~0 (>80 Å, Fig. 3c; we observed Cy5 fluorescence upon direct 640 nm excitation in all traces selected, confirming the presence of

acceptor at a distance from donor beyond FRET). Most molecules showed stable $E_{FRET}$ states with only ~4% of them undergoing slow transitions between the two states (Fig. 3d). When the Mg$^{2+}$ concentration was increased to 20 mM, the population of the major FRET state increased from ~61 to ~88%, now with a slightly higher $E_{FRET}$ value of $0.40 \pm 0.12$ (~58 Å); it saturated at ~89% with an $E_{FRET}$ of $0.41 \pm 0.12$ at 50 mM Mg$^{2+}$ (Fig. 3c). These observations demonstrate that the tRNA:T-box core complex exists in two distinct conformations that interconvert slowly, with the more compact structure favored by high Mg$^{2+}$ concentrations. Notably, an RNA kink turn (K-turn)[26,27] motif exists below the Specifier loop of Stem-I in the T-box core. This motif is known to be bi-stable, kinking with a similarly high Mg$^{2+}$ dependence[28], a motion consistent with the observed

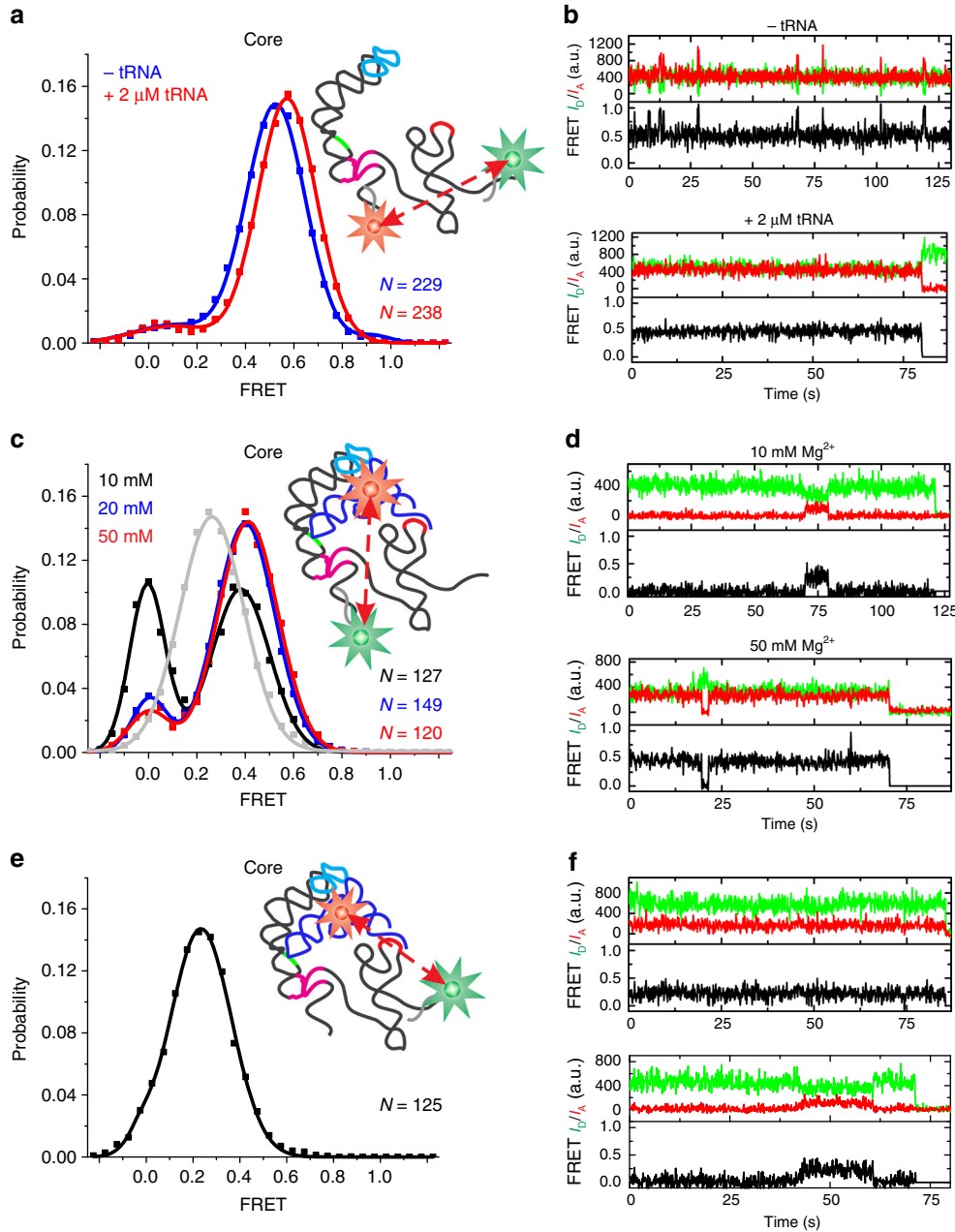

**Fig. 3** smFRET analysis of tRNA binding to core T-box riboswitch. **a** FRET histograms and **b** exemplary traces for an smFRET measurement between the base of Stem-I and the base of antiterminator, in the presence and absence of unlabeled tRNA. **c** Histograms of smFRET measurements between tRNA and the base of Stem-I under different $Mg^{2+}$ concentrations. The gray curve shows the corresponding FRET histogram for the isolated Stem-I construct. **d** Example smFRET traces for measurement in **c**. **e** Histogram of smFRET measurement between tRNA and the base of antiterminator. **f** Example smFRET traces for measurement in **e**

shortening of the Stem-I base-to-tRNA elbow vector upon $Mg^{2+}$ titration.

The 3′-end of tRNA needs to interact with the T-box sequence to stabilize the antiterminator and prevent transcription termination (Fig. 1a, b). To investigate the proximity of bound tRNA to the antiterminator and to visualize the kinetics of this key interaction, we measured FRET from the base of the antiterminator to the tRNA variable loop. A FRET histogram of the T-box core:tRNA complex displayed a dominant (~97%) state with $E_{FRET} = 0.23 \pm 0.13$ and a very minor (3%) state with $E_{FRET} \sim 0$ (Fig. 3e), corresponding to distances of ~65 and >80 Å, respectively. Visual inspection of individual smFRET traces showed that transitions between these two states are rare (<5% molecules), suggesting slow dynamics (Fig. 3f). Taken together,

our smFRET analyses support a model wherein the *glyQS* T-box riboswitch is largely pre-folded, with a K-turn hinge that encodes a bi-stable contractive motion along a specific (Stem-I base-to-tRNA elbow) axis.

**Antiterminator anchors tRNA, conferring ultra-high stability.** Interaction of the tRNA 3′-end with the T-box in the antiterminator is thought to stabilize the structure of the complex that results in transcriptional read-through[3-5]. To understand the contribution of this key interaction, we measured the tRNA binding kinetics to the T-box core riboswitch (Fig. 4a). Individual smFRET time traces showed largely static heterogeneity, displaying either short or long binding events, although a small

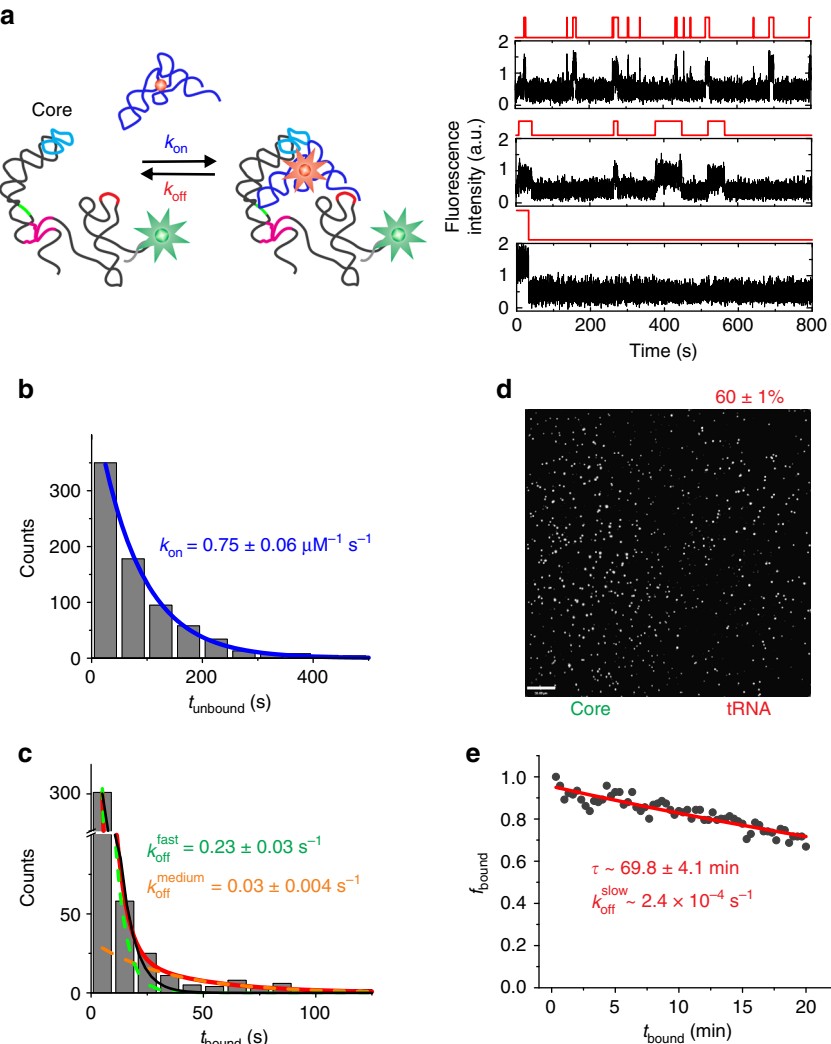

**Fig. 4** Kinetic analysis of tRNA binding to core T-box riboswitch. **a** Schematic and representative traces of tRNA binding to the core T-box riboswitch at 10 mM $Mg^{2+}$ showing static heterogeneity between different traces. **b** Dwell-time distribution of $t_{unbound}$ fit with a single-exponential function (blue). **c** Dwell-time distribution of $t_{bound}$ fit with a double-exponential function (red), deconvolved into the $k_{off}^{fast}$ (green) and $k_{off}^{medium}$ (orange) contributions. An alternative fit with a single-exponential function is shown as a black curve. **d** Average image of first 30 frames (3 s) for uncharged tRNA binding to the core riboswitch. The spots in the Cy5 channel correspond to tRNA molecules in the ultra-stable complexes that remained associated with core T-box ~10 min after immobilization to the slide. **e** Fraction of the surface-bound tRNA molecules co-localized with the core T-box riboswitch, $f_{bound}$, as a function of time. The data are fit with a single-exponential function to determine the tRNA bound lifetime of the ultra-stable complexes. Errors are s.d. of three independent replicates

fraction of (~15–20%) molecules displayed both (Fig. 4a). Notably, the tRNA association kinetics was homogeneous with a $k_{on}$ of $0.75 \pm 0.06 \times 10^{6} \, M^{-1} \, s^{-1}$ ($\tau_{unbound}$ ~79 ± 2 s at 16 nM tRNA; Fig. 4b), within error the same as the Stem-I construct. The core riboswitch construct bound tRNA tightly, with heterogeneous (bi-exponential) kinetics; the observable dissociation events yielded $k_{off}^{fast} = 0.23 \pm 0.03 \, s^{-1}$ ($\tau_{bound} = 4.3 \pm 0.6$ s; 72 ± 6%) and $k_{off}^{medium} = 0.03 \pm 0.004 \, s^{-1}$ ($\tau_{bound} = 34.2 \pm 4.6$ s; 28 ± 6%; Fig. 4c). While $k_{off}^{fast}$ for the T-box core is similar to the $k_{off}$ for Stem-I only, suggesting similar interactions, the $k_{off}^{medium}$ is ~8-fold slower and thus most likely includes an interaction involving the 3′-end of tRNA with the T-box antiterminator bulge. These findings support a model wherein the antiterminator does not contribute to the initial recruitment of tRNA, but significantly stabilizes the complex once formed.

Our observation using smFRET of very slow tRNA:T-box core dissociation events, even exceeding 1 h, suggested that an additional dissociation rate constant exists beyond the observable

$k_{off}^{medium}$. This third population of particularly long-lived (ultra-stable) complexes required us to measure their tRNA dissociation rate constants by immobilizing complexes on the slide and monitoring the disappearance of tRNA over an extended period of time. Molecules were imaged intermittently (once every 20 s) with low laser power to minimize photobleaching. Using single-molecule colocalization, we found that 60 ± 1% of all T-box core molecules retained tRNA after 10 min (Fig. 4d), suggesting that they form ultra-stable complexes involving antiterminator interactions, with a bound tRNA lifetime measured to be 69.8 ± 4.1 min, or a very slow $k_{off}^{slow} < 2.4 \times 10^{-4} \, s^{-1}$ (Fig. 4e). Extrapolating this rate to time 0 results in an estimate that ~69% of all binding competent T-box core molecules form the ultra-stable complexes under high (0.5 μM), and stoichiometric, tRNA concentrations. Notably, tRNA was not observed to bind the individual antiterminator stem loop, Specifier sequence, or DTM platform structural elements (Supplementary Fig. 2e). Taken together, these data suggest that, while Stem-I alone governs the

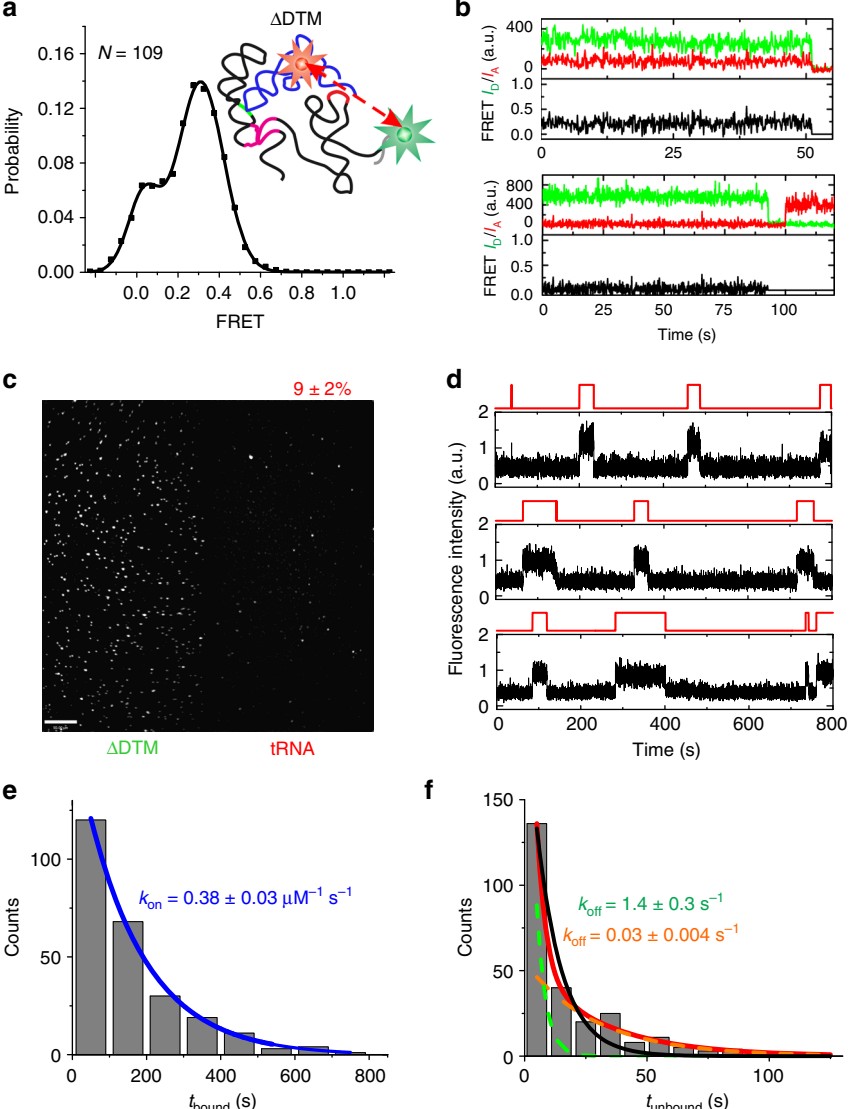

**Fig. 5** smFRET and kinetic analysis of tRNA binding to ΔDTM construct. **a** FRET histogram and **b** exemplary traces for smFRET measurement between tRNA and the base of antiterminator in the ΔDTM construct at 50 mM $Mg^{2+}$. **c** Average image of first 30 frames (3 s) for uncharged tRNA binding to the ΔDTM construct at 10 mM $Mg^{2+}$. The few spots in the Cy5 channel represent tRNA molecules that remained associated with the ΔDTM T-box construct ~3 min after immobilization to the slide. **d** Representative traces for tRNA binding to the ΔDTM construct at 10 mM $Mg^{2+}$. **e** Dwell-time distribution of $t_{unbound}$ fit with single-exponential function (blue) to estimate $k_{on}$. **f** Dwell-time distribution of $t_{bound}$ fit with a double-exponential function (red), deconvolved into the fits for the fast (green) and slow (orange) components. An alternative fit with a single-exponential function is shown in black. Errors are s.d. of at least two independent replicates

homogeneous binding kinetics, dissociation is fast if Stem-I alone interacts with the tRNA through its Specifier and DTM platform, and slows once the antiterminator can also be engaged, giving rise to two slower dissociation rate constants.

**DTM contributes to faster $k_{on}$ and to form ultra-stable complexes.** To further examine the role of the DTM in tRNA binding, we used a ΔDTM T-box core variant lacking the motif at the distal end of Stem-I. Measuring FRET from the base of the antiterminator to the tRNA variable loop, we observed complexes form in standard buffer; however, they were short-lived and mostly dissociated within a minute, much faster than those of the wild-type core. This demonstrates that the ΔDTM variant can bind tRNA, but less tightly than the T-box core construct. To extend the tRNA residence times for smFRET studies, we increased the concentration of $Mg^{2+}$ from 10 to 50 mM. The

resulting FRET histogram displayed a major (~78%) state with $E_{FRET} = 0.31 \pm 0.11$ and a minor (~22%) state with $E_{FRET}$ ~0, corresponding to distances of ~62 and >80 Å, respectively (Fig. 5a) with only rare transitions between them, analogous to the riboswitch core:tRNA complexes (Fig. 5b). The FRET distributions of the ΔDTM:tRNA and core:tRNA complexes (Fig. 3e) suggests that the tRNA maintains a similar orientation relative to the antiterminator in the absence of the DTM:elbow interaction, although the probability to adopt a long-distance (>80 Å) conformation increases from 3 to 22%.

We next probed the function of the DTM on the tRNA binding kinetics in standard buffer. As determined during our smFRET measurements, tRNA binds to the ΔDTM variant unstably so that only a small fraction, ~9% of tRNA, remained bound after 3 min (Fig. 5c). The complexes completely dissociated within 10 min, in contrast to the native core T-box riboswitch, demonstrating that the DTM:tRNA elbow interaction is involved in complex

formation. The binding kinetics, as determined by single-molecule fluorescence colocalization, are homogeneous (Fig. 5d), with a $k_{on}$ of $0.38 \pm 0.03 \times 10^6$ M$^{-1}$ s$^{-1}$ ($\tau_{unbound} = 165 \pm 16$ s at 16 nM tRNA) (Fig. 5e). The $k_{on}$ value is only half of that measured for the native T-box core and suggests that the DTM plays a role in fast tRNA binding. Dissociation displays biphasic kinetics with a fast $k_{off} = 1.4 \pm 0.3$ s$^{-1}$ ($\tau_{bound} = 0.7 \pm 0.3$ s; 27 ± 6%) minor component and a slower $k_{off} = 0.03 \pm 0.004$ s$^{-1}$ ($\tau_{bound} = 29.4 \pm 3.5$ s; 73 ± 6%) major component (Fig. 5f). Remarkably, the slower $k_{off}$ for the ΔDTM variant is very similar to $k_{off}^{medium}$ of the wild-type T-box core, indicating that $k_{off}^{medium}$ most likely represents wild-type complexes lacking the DTM: tRNA elbow interaction.

**tRNA aminoacylation leads to loss of ultra-stable complexes.** To examine how aminoacylation alters the tRNA binding kinetics, we used Gly-tRNA$^{Gly}$ in our single-molecule fluorescence coincidence assays. Gly-tRNA$^{Gly}$ has a longer chemical half-life than many other aa-tRNAs[29] and is relatively stable at room temperature and pH 7.0[30]. The stability of Gly-tRNA$^{Gly}$ is expected to be further enhanced under the acidic pH of 6.1 of our binding experiments. Aminoacylation using *Escherichia coli* S100 extract yielded charging efficiencies of ~90%, as determined by denaturing acid-urea gel electrophoresis (Fig. 6a). The single-molecule fluorescence time traces of Gly-tRNA$^{Gly}$ binding to Stem-I exhibited short binding events, similar to binding of uncharged tRNA to Stem-I (Fig. 6b). We measured homogeneous values of $k_{on} = 0.74 \pm 0.03 \times 10^6$ M$^{-1}$ s$^{-1}$ and $k_{off} = 0.18 \pm 0.02$ s$^{-1}$ (Fig. 6b and Supplementary Figs. 5a, b), which are very close to the rate constants obtained for the uncharged tRNA. These data

demonstrate that aminoacylation of the tRNA has no significant effect on the kinetics of binding to Stem-I.

For the T-box core, the single-molecule fluorescence coincidence traces for Gly-tRNA$^{Gly}$ (Fig. 6c) revealed a homogeneous $k_{on} = 0.75 \pm 0.06 \times 10^6$ M$^{-1}$ s$^{-1}$ (Supplementary Fig. 5c), identical to the value for uncharged tRNA. Dissociation kinetics were again biphasic, with $k_{off}^{fast} = 0.19 \pm 0.03$ s$^{-1}$ ($\tau_{bound}$ ~5.3 ± 0.5 s; 46 ± 8%) and $k_{off}^{medium} = 0.04 \pm 0.002$ s$^{-1}$ ($\tau_{bound}$ ~24.5 ± 1 s; 54 ± 6%) (Supplementary Fig. 5d). These results show that $k_{off}^{fast}$ is unaffected by tRNA charging, whereas $k_{off}^{medium}$ slightly increases from ~0.03 to ~0.04 s$^{-1}$ (corresponding to a decrease in $\tau_{bound}$ from ~34.2 to ~24.5 s). The most significant difference observed between experiments performed using charged and uncharged tRNA is that only ~17% of all tRNAs in the charged tRNA experiments remain bound to the slide after 5 min, whereas 61% of tRNAs remain bound in the uncharged tRNA experiments (Fig. 6d). The residual charged tRNA remaining bound likely reflects, the minor fraction of uncharged tRNA present in this Gly-tRNA$^{Gly}$ preparation. Importantly, these results indicate that the presence of even a small amino acid like Gly on the tRNA 3′-end suppresses the formation of the ultra-stable T-box core complexes.

**EF-Tu does not alter the binding kinetics of Gly-tRNA$^{Gly}$.** In vivo, aminoacylated tRNAs exist primarily in a ternary complex bound to EF-Tu:GTP. To study whether the binding kinetics of the charged tRNA to the riboswitch are affected by complexation with EF-Tu, we probed the interaction of the ternary complex with the T-box riboswitch (Fig. 7). As a control, we first confirmed that the Cy5 label on the tRNA allows for the formation of this complex. Using an electrophoretic mobility shift assay we

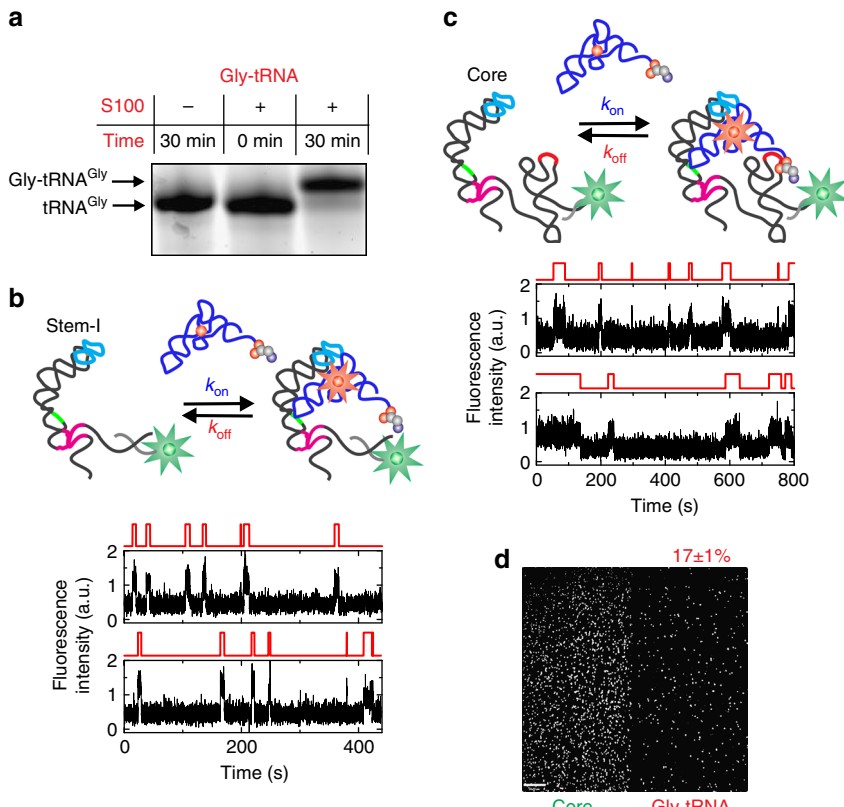

**Fig. 6** Effect of aminoacylation on the tRNA binding kinetics. **a** Acid-urea PAGE showing high-efficiency of tRNA aminoacylation using *E. coli* S100 extracts. Schematic and representative single-molecule traces showing Gly-tRNA$^{Gly}$ binding to **b** Stem-I and **c** the core T-box riboswitch. **d** Average image of first 30 frames (3 s) for Gly-tRNA$^{Gly}$ binding to the core T-box construct. Approximately 17% of Gly-tRNA$^{Gly}$ molecules are in ultra-stable complexes

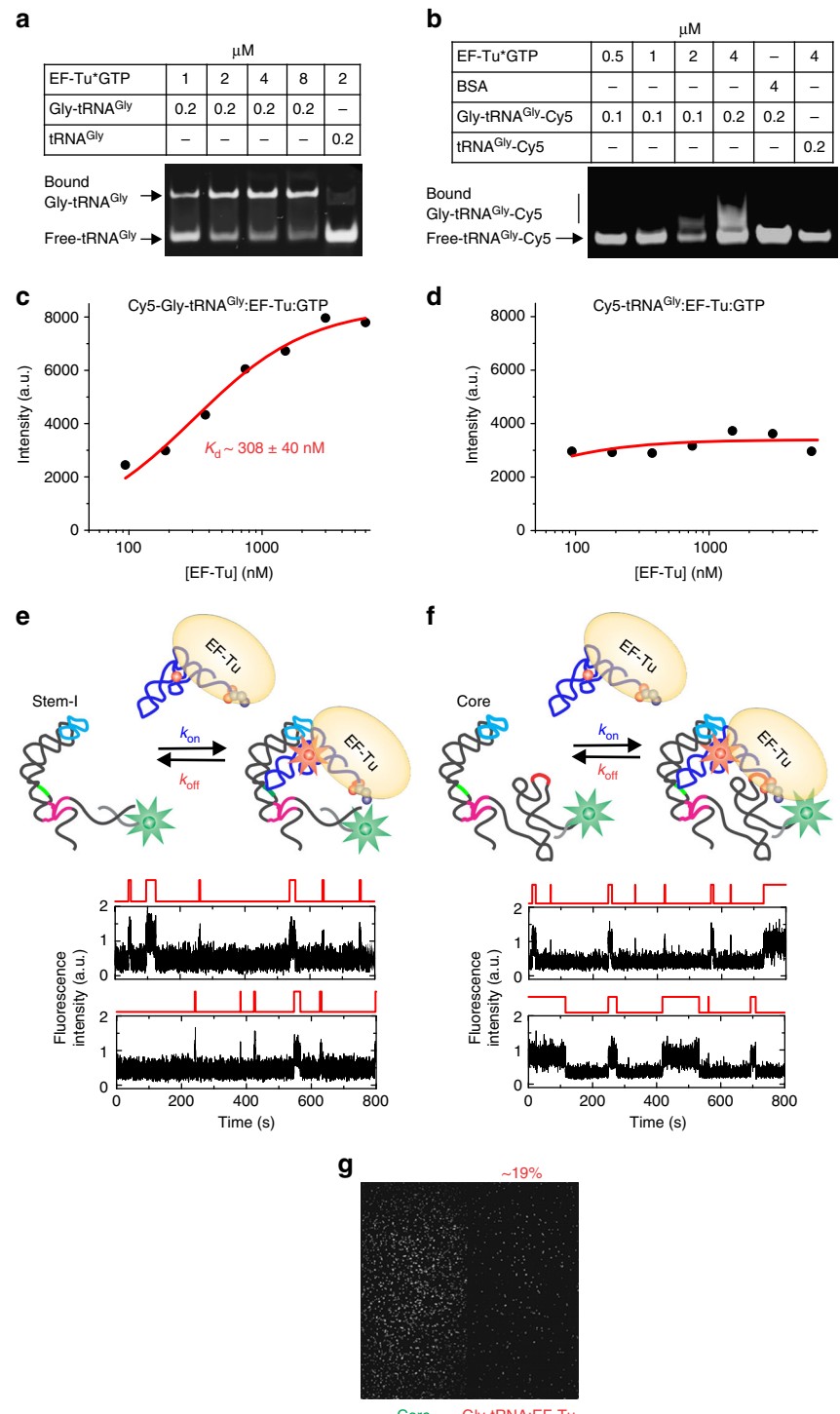

**Fig. 7** Kinetic analysis of ternary complex binding to the T-box riboswitch. Native PAGE showing mobility shift of **a** unlabeled and **b** Cy5-labeled Gly-tRNA$^{Gly}$ binding to EF-Tu. Estimation of apparent $K_d$ for EF-Tu binding to **c** Cy5-Gly-tRNA$^{Gly}$ and **d** Cy5-tRNA$^{Gly}$ using RNase protection assays with fits to non-cooperative ($n = 1$) Hill equations. Schematic and representative single-molecule traces showing Gly-tRNA$^{Gly}$:EF-Tu:GTP binding to **e** Stem-I and **f** the core T-box riboswitch. **g** Average image of the first 30 frames (3 s) for Gly-tRNA$^{Gly}$:EF-Tu:GTP binding to the core T-box construct

found the transcribed Gly-tRNA$^{Gly}$ and U46 Cy5-labeled Gly-tRNA$^{Gly}$ to bind EF-Tu:GTP with similar apparent affinities (~1.5 µM; Fig. 7a, b and Supplementary Fig. 5e), comparable to previously measured affinities based on a similar assay[31]. As expected, the ternary complex was not observed for non-aminoacylated tRNA$^{Gly}$ or Cy5-tRNA$^{Gly}$ (Fig. 7a, b), confirming the specificity of EF-Tu:GTP for aminoacylated tRNA. Measuring instead the affinity through the protection of Cy5-Gly-tRNA$^{Gly}$ by EF-Tu:GTP against RNase degradation followed by nitrocellulose binding[32,33] yielded a more realistic, higher-affinity apparent $K_d$ of $310 \pm 40$ nM (Fig. 7c), establishing the fraction of Cy5-Gly-tRNA$^{Gly}$ bound to EF-Tu:GTP as >80% under the conditions of our single-molecule fluorescence coincidence assays. As expected, EF-Tu did not protect the uncharged tRNA against RNase degradation, further confirming the specific binding to the Cy5-Gly-tRNA$^{Gly}$ (Fig. 7d).

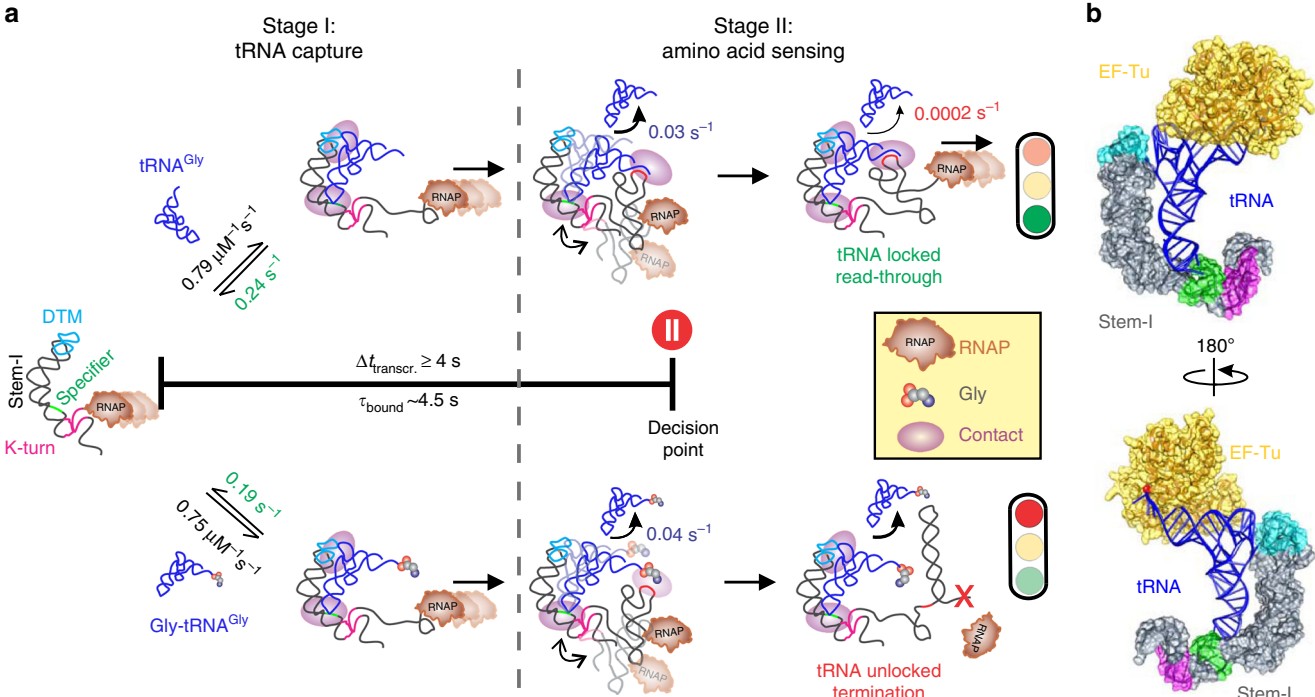

**Fig. 8** Kinetic model of aminoacylation-dependent transcription control by the *glyQS* T-box riboswitch. **a** Binding and capture of tRNA begins once Stem-I, with the DTM and the Specifier sequence, emerges from the RNA polymerase (RNAP). Stem-I binds both the charged and uncharged tRNAs with similar $k_{on}$ and $k_{off}$. The average bound dwell-time of ~4.5 s is close to the time required for the RNAP to reach the decision point, immediately prior to formation of the terminator where a strong pause site (at position 178, shown using a red pause symbol) is present. Sensing of the aminoacylation state begins with the antiterminator which the bi-stable K-turn positions close to the tRNA 3′-end. The tRNA elbow:DTM interaction can be released in these complexes, leading to dissociation of uncharged tRNA with a $k_{off}^{medium}$ ~ 0.03 s$^{-1}$. Aminoacylation still permits tRNA 3′-end interaction with the antiterminator loop, albeit with a slightly faster $k_{off}^{medium}$ ~ 0.04 s$^{-1}$. Formation of all the three contacts, however, results in "snap locking" of the uncharged tRNA into an "ultra-stable" ring-like commitment complex characterized by a very slow $k_{off}$ ~2 × 10$^{-4}$ s$^{-1}$ that allows transcription read-through. Notably, even the small amino acid on Gly-tRNA$^{Gly}$ specifically abrogates these complexes, thus terminating transcription. **b** Structural model of the Gly-tRNA$^{Gly}$:EF-Tu:GTP ternary complex bound to T-box riboswitch Stem-I, showing that the Van der Waals surfaces of EF-Tu and Stem-I have no sterical clash while the tRNA 3′-end remains accessible

Employing Stem-I in our coincidence assay, we measured homogeneous kinetics with $k_{on} = 0.65 ± 0.03 × 10^6$ M$^{-1}$ s$^{-1}$ and $k_{off} = 0.18 ± 0.01$ s$^{-1}$ (Fig. 7e and Supplementary Fig. 5f, g), indicating that EF-Tu does not significantly affect binding kinetics to Stem-I compared to free Gly-tRNA$^{Gly}$. For the T-box core molecule, we determined a very similar $k_{on} = 0.76 ± 0.07 × 10^6$ M$^{-1}$ s$^{-1}$, while $k_{off}$ displayed biphasic kinetics with $k_{off}^{fast} = 0.25 ± 0.04$ s$^{-1}$ ($\tau_{bound}$ ~4.0 ± 0.6 s; 55 ± 5%) and $k_{off}^{medium} = 0.03 ± 0.008$ s$^{-1}$ ($\tau_{bound}$ ~24.5 ± 1 s; 45 ± 5%) (Fig. 7f and Supplementary Fig. 5h, i). Only a small fraction (~19%) of tRNA remains stably bound to the core riboswitch after 5 min (Fig. 7g) when probed for the ability to form ultra-stable complexes, consistent with the fraction of uncharged tRNA remaining in our Gly-tRNA$^{Gly}$ preparation. Together, these data demonstrate that neither Gly-tRNA$^{Gly}$ nor the Gly-tRNA$^{Gly}$:EF-Tu:GTP ternary complex form ultra-stable complexes with the T-box riboswitch and that complexation with EF-Tu has only little effect on tRNA binding and dissociation, compared to uncharged tRNA.

## Discussion

Despite the discovery of the T-box riboswitch over 20 years ago[6], its molecular switching mechanism remains poorly understood. The recent studies of Stem-I:tRNA complexes[15,16] provide detailed structural information; however, these complexes lack the antiterminator and thus insights into the amino acid sensing mechanism[3–5]. To elucidate this mechanism, we have used smFRET and a single-molecule fluorescence coincidence assay to probe the kinetic and architectural underpinnings of the *glyQS*

T-box riboswitch interaction with its cognate ligand, tRNA$^{Gly}$ (Fig. 8a). Remarkably, while we found Stem-I alone to rapidly bind tRNA, consistent with a caliper-like interaction with the tRNA using the Specifier sequence and DTM, it suffers from a fast dissociation rate constant (Fig. 8a) that was not gleaned from previous equilibrium affinity measurements and crystallization[13,14]. We further observed that the T-box core adopts two alternate, Mg$^{2+}$-dependent, pre-organized conformations characterized by medium and large separations between the bound tRNA's elbow and the base of Stem-I, consistent with the presence of the bi-stable K-turn motif embedded at the base of the stem[26–28]. Upon kinking the RNA backbone by ~120°, the tRNA acceptor stem will be positioned closer to the antiterminator, as necessary for amino acid sensing (Fig. 8a). Comparison of the corresponding distances (~64 Å between the base of Stem-I and bound tRNA variable loop) in the kinked conformer of the K-turn from the crystal structure[15] with the extended conformation observed using NMR spectroscopy (with a corresponding between the distance of >90 Å) in a proximal portion of an apo *tyrS* T-box Stem-I[26] supports assignment of the observed ~0.4 and ~0 FRET states to the kinked and extended conformations, respectively.

Riboswitches functioning at the level of transcription termination, such as the *glyQS* T-box riboswitch, are generally thought to be controlled kinetically[19,20] so that the rate of ligand binding —as uniquely measured here for two T-box transcription intermediates—is crucial to the outcome of gene expression. The $k_{on}$ and $k_{off}$ values of the T-box riboswitch core parallel the ligand binding (~10$^4$ to 10$^6$ M$^{-1}$ s$^{-1}$) and dissociation rate constants

($10^{-1}$ to $10^{-4}$ s$^{-1}$) of metabolite-binding riboswitches[34], suggesting that similar ligand recognition kinetics apply to both large and small ligands. Our determination of a fast binding rate constant $k_{on}$ of $\sim 0.8 \times 10^6$ M$^{-1}$ s$^{-1}$ for the *glyQS* T-box is in line with the expectations for a co-transcriptional regulator of gene expression. Notably, similar $k_{on}$ values for the Stem-I and the T-box core show that the antiterminator and its single-stranded linker do not participate in initial tRNA capture (Fig. 8a). The dissociation rate constant $k_{off}$ of $\sim 0.2$ s$^{-1}$ further demonstrates that Stem-I alone cannot retain the tRNA for an extended period, independent of its aminoacylation state (Fig. 8a). In contrast, tRNA dissociation from the T-box core displays two additional, slower rate constants: $k_{off}^{medium}$ of $\sim 0.03$ s$^{-1}$, which is only slightly accelerated to $\sim 0.04$ s$^{-1}$ upon tRNA aminoacylation, and $k_{off}^{slow}$ of $\sim 2.4 \times 10^{-4}$ s$^{-1}$ that results in an ultra-stable complex (Fig. 8a). These observations demonstrate that the downstream antiterminator sequence is required to slow the dissociation of tRNA. Deleting the DTM from Stem-I of the T-box core decreases $k_{on}$ by $\sim 2$-fold, but has a negligible effect on $k_{off}^{medium}$, indicating that the DTM platform—conserved among transcriptionally but not translationally acting T-box riboswitches[35,36]—plays an important role in fast capturing of ligand. Our tRNA binding affinities obtained from the single-molecule binding experiments ($K_d$ ~50 nM and ~300 nM for the riboswitch core and Stem-I, respectively) compare favorably with previous tRNA binding measurements performed using ITC on various *glyQS* T-box riboswitch variants[15,25]. Taken together, our kinetic and architectural analyses lead to the mechanistic model in Fig. 8a wherein the three T-box:tRNA contacts—DTM:elbow, Specifier sequence: anticodon, and the antiterminator:acceptor stem—are engaged hierarchically: The DTM and Specifier sequence of Stem-I capture the tRNA body, followed by amino acid sensing through exploration of the antiterminator:acceptor stem interaction; only in the absence of an amino acid can the tRNA "snap lock" into an ultra-stable, three-contact complex that stabilizes sufficiently for transcription read-through of the otherwise metastable antiterminator.

In the Stem-I:tRNA crystal structures, the tRNA elbow stacks against the DTM of Stem-I[15,16]. Recently, two SAXS studies proposed models of the *B. subtilis glyQS* core T-box:tRNA complexes that differed in the presence or absence of the tRNA elbow: DTM contact in the final structure, suggesting conformational flexibility[25,37]. While one study suggested the presence of all three contacts in the final complex[37], the other study (using an RNA similar to the T-box core in the current work) found that the interaction between the tRNA elbow and the platform was largely missing in the T-box core:tRNA complex, perhaps because it serves as an intermediate toward the formation of a final complex involving the downstream antiterminator[25]. Our detection of similar $k_{off}^{medium}$ values for tRNA binding to the wild-type and ΔDTM mutant T-box cores supports the notion that the tRNA elbow–DTM interaction is at least partially lost after the initial tRNA capture, perhaps to test the strength of the antiterminator: acceptor stem interaction (Fig. 8a). By contrast, the T-box core molecules that form the ultra-stable complexes require the DTM, leading to cooperation between the three T-box:tRNA contacts as they lock into a stable, ring-like structure (Fig. 8a).

Analogous to the ribosome[38] and the spliceosome[39], the ultra-stable, locked T-box:tRNA complex represents a "commitment" complex that is preceded by assemblies involved in kinetic proofreading of the aminoacyl-tRNA on a time scale dictated by transcription elongation. Using charged tRNA, we have shown that the presence of Gly only slightly affects the $k_{off}^{medium}$, suggesting that the aminoacylated tRNA 3′-end can still weakly interact with the T-box sequence in the antiterminator (Fig. 8a). However, the dramatic loss of the commitment complex (i.e.,

$k_{off}^{slow}$) upon charging demonstrates the importance of the hierarchical assembly process and reveals the exquisite sensitivity of the snap-locking mechanism in sensing even a small chemical modification. Additionally, our kinetic measurements showed that dissociation of Gly-tRNA$^{Gly}$ from the T-box core is not significantly altered when it is part of the tRNA:EF-Tu:GTP ternary complex. This observation is consistent with a structural model assembled using the individual Stem-I:tRNA[15] and Phe-tRNA$^{Phe}$:EF-Tu:GTP[40] structures that shows no steric clash between the Stem-I and EF-Tu (Fig. 8b). Furthermore, our findings lend support to the idea that the T-box riboswitch may have evolved to sense tRNA aminoacylation status prior to the appearance of EF-Tu as a translational factor[30].

Previous in vitro kinetic studies on the *glyQS* T-box riboswitch have identified three major RNA polymerase (RNAP) pause sites[18]. One of those sites includes a uridine nucleotide stretch at position 138 (Fig. 1b) in Stem-III loop with a long half-life of ~3 min at low NTP concentration. However, under high intracellular NTP concentrations and in the presence of cellular transcription-modulating factors, this long pause is absent, as evidenced by the high-throughput NET-seq data from Larson et al.[41], while a relatively long pause is still detected at residue 178 (Supplementary Fig. 6). This pause would position the T-box sequence just outside the polymerase exit channel (~15 residues upstream). We propose that the pause at this location will allow time for the T-box sequence to interrogate the aminoacylation status of the bound tRNA, relying on the base pairing interaction with the tRNA 3′-end. The base pairing would then promote folding of the antiterminator once the pause ends, before the rest of the terminator sequence is made. Assuming an average rate of transcription by bacterial RNAP of ~20–25 nucleotides per second[20,42], in the absence of any RNAP pausing before position 178, only a short (~4 s) time window would be available for Stem-I to capture tRNA and stabilize the weak antiterminator before the RNAP reaches this key decision point. Remarkably, this time is close to the bound time ($\tau_{bound} = 1/k_{off}$) of tRNA on Stem-I of ~4.5 s (Fig. 6a). Thus, our kinetic measurements support the notion that the T-box riboswitch may be fine-tuned for kinetic control of transcription, providing through polymerase pausing the minimal time needed for the tRNA to remain bound and be sensed for its aminoacylation state. A similar mode of action has been proposed for certain transcriptionally acting metabolite-binding riboswitches[20,43].

Our work highlights the importance of single-molecule measurements in unraveling the complexity in RNA–RNA interactions, where transient and heterogeneous binding and dissociation kinetics are key to understanding their mechanism. Our work provides a quantitative spatiotemporal analysis of T-box riboswitch actuation, suggesting a complex hierarchical mode of tRNA recognition and amino acid sensing that may prove to be a model for other regulatory systems and may lay the foundation for future antibiotic drug targeting approaches. Finally, the complementation of the U-shaped T-box riboswitch by the L-shaped tRNA to snap into an ultra-stable ring-like commitment complex may inspire novel biomimetic architectures in the emerging field of RNA nanotechnology[44–46].

## Methods

**Transcription and native purification of *glyQS* T-box riboswitch**. The *glyQS* T-box riboswitch RNAs were made by in vitro transcription of PCR-generated DNA templates using AmpliScribe™ T7 High Yield Transcription Kit (Illumina Inc.). Transcription reactions were done using 3 mM 5′-biotin-GMP (Trilink Bio-Technologies), 2 mM GTP, and 5 mM each of ATP, CTP, and UTP, and then incubated for 4–6 h[47]. At the end of the reaction, precipitate in the transcription reaction was removed by centrifugation. The transcribed RNA was purified from the excess free nucleotides and salts by filtration using a 10 kDa MW cutoff filter, and concentrated and reconstituted with 10 mM Tris-HCl, pH 7.4, 50 mM KCl, 2

mM $MgCl_2$. The concentration of RNA was measured by UV-Vis spectroscopy and its purity was analyzed using denaturing polyacrylamide gel electrophoresis (PAGE). The resulting natively purified RNA contained a mixture of different length fragments, with a major full-length product. Only the full-length T-box riboswitch variant molecules were visualized using a complementary Cy3-labeled DNA oligonucleotide (Supplementary Fig. 1) that hybridizes to the 3′-end of the RNA.

**Fluorophore labeling of DNA/LNA oligonucleotides**. The 15-nucleotide LNA/DNA oligonucleotide with 5′-amine linker (/5AmMC6/ +T+GTTCT+GT +TGATC+C+C, RNA $T_m$ ~63 °C, where "+N" represents an LNA nucleotide, Supplementary Fig. 1) was chemically synthesized by Exiqon and labeled with Cy3 or Cy5. One dye pack of Cy3/Cy5 monofunctional NHS ester (GE Healthcare) was dissolved in 30 µl dimethyl sulfoxide and used to label ~5 nmol of the LNA oligonucleotide in a 50 µl reaction volume containing 0.1 M sodium bicarbonate buffer, pH 8.7. The reaction was incubated at room temperature (RT) for 4 h while tumbling in the dark. Excess unlabeled dye was removed by precipitation with three volumes of 100% ethanol and 300 mM sodium acetate, pH 5.2. After centrifugation, the pellet was dried in a SpeedVac concentrator and suspended in autoclaved milliQ water for further use. The 14-nucleotide DNA oligonucleotide (5′-TGTTCGTGGTGCTC-Cy3-3′) complementary to the 3′-end of the T-box was chemically synthesized with a 3′-Cy3 by Integrated DNA Technologies (IDT).

**Creation of Cy5-labeled $tRNA^{Gly}$**. The Cy5-labeled $tRNA^{Gly}$ RNA was made using a two-piece ligation strategy[48]. A 35-nucleotide 5′-half of the tRNA and the 40-nucleotide 3′-half with an internal Cy5 label incorporated between U46 and C47 were chemically synthesized from IDT. An equimolar mixture of the two RNAs was annealed by heating at 95 °C for 1 min followed by cooling on ice. The RNAs were then ligated by using 1 U/µl concentration of T4 RNA Ligase 1 (New England Biolabs) in the supplied 1x manufacturer buffer for 1 h at 37 °C. RNA concentration was ~20–70 µM in the ligation reaction. The final ligated product corresponding to the full-length tRNA was purified using denaturing 8 M urea PAGE in 1x TBE (Tris/Borate/EDTA). The purified tRNA was folded by heating at 65 °C for 3 min followed by slow cooling at RT for 20 min.

**Formation of T-box riboswitch:tRNA complexes for smFRET**. One micromole of the T-box riboswitch RNA was first incubated with 10 µM of DNA or 5 µM of LNA oligonucleotide for 15 min at room temperature in standard buffer (10 mM Tris-HCl, pH 6.1, 50 mM KCl, 10 mM $Mg^{2+}$). This mixture was then used to make a stock of 0.5 µM T-box:tRNA complex by incubating with 1 µM of Cy5-labeled $tRNA^{Gly}$ at RT for at least 15 min. For smFRET measurements of the weakly binding ΔDTM construct, the complexes were formed on ice while increasing the $Mg^{2+}$ concentration to 50 mM. Required dilutions of the product to 20–50 pM were made in standard buffer for immobilization onto the slide for smFRET experiments.

**Aminoacylation of $tRNA^{Gly}$ and acid-urea PAGE**. Aminoacylation of Cy5-labeled $tRNA^{Gly}$ was achieved using E. coli S100 extracts (tRNA probes, College Station, TX, USA) in a 25 µl reaction volume following a protocol previously described[49]. Briefly, ~2 µM of Cy5-$tRNA^{Gly}$ was charged with 3 µl of E. coli S100 extract in a 50 µl reaction containing 100 mM HEPES-KOH, pH 7.6, 200 µM glycine, 10 mM ATP, 1 mM dithiothreitol, 20 mM KCl, 20 mM $MgCl_2$, and incubated at 37 °C for 30 min. The reaction was stopped by adding 3 M sodium acetate, pH 5.2, to a final concentration of 100 mM in 100 µl volume. Aminoacylated tRNA was purified by performing phenol–chloroform extraction to remove the proteins using phase-lock gel tubes. The concentration of the Gly-$tRNA^{Gly}$ was measured using a UV-Vis spectrophotometer and multiple aliquots of it were made and stored at −80 °C. An acid-urea PAGE[49] was used for checking the aminoacylation efficiency of the reaction. We added 2 µl of the 25 µl charging reaction to 98 µl of gel loading buffer (0.1 M sodium acetate, pH 5.0, 8 M urea, 0.05% bromophenol blue, 0.05% xylene cyanol FF). Ten microliters of this mixture was run on a 10% acid, 8 M urea, gel using 10 W power at 4 °C for ~20–25 h between 30-cm-long gel plates. The gel was cast using 0.1 M sodium acetate, pH 5.0, and the same buffer was used for gel electrophoresis. The buffer was constantly circulated between the top and bottom chambers using a peristaltic pump to maintain the pH at 5.0. The gel was scanned with a Typhoon 9410 Variable Mode Imager (GE Healthcare Life Sciences) with ImageQuant software (Molecular Dynamics) using a 640 nm red laser for imaging Cy5-$tRNA^{Gly}$. Aminoacylated tRNA will move slightly slower than uncharged tRNA, enabling the estimation of charging efficiency. Under our reaction conditions, the aminoacylation efficiency was close to ~90% (Fig. 5a). For the single-molecule binding experiments, Cy5-labeled Gly-$tRNA^{Gly}$ was diluted to the required concentration in standard buffer.

**Formation of ternary complexes containing EF-Tu:GTP**. EF-Tu was purchased from tRNA probes (tRNAprobes.com) and was stored at −20 °C, in 30% glycerol, and at a concentration of 180 µM. Unlabeled $tRNA^{Gly}$ was produced by in vitro transcription and gel purified before charging with glycine. Charging of EF-Tu with GTP was performed based on published protocols[31] and was prepared freshly for each experiment, under the following conditions: 70 mM $NH_4Cl$, 7 mM $MgCl_2$, 30

mM KCl; and 50 mM Tris-acetate, pH 7, 5 mM β-mercaptoethanol (BME), 12 µM EF-Tu, 8 mM GTP, 5 U/ml pyruvate kinase from rabbit muscle (Sigma Aldrich), and 7 mM phosphoenolpyruvate. The mixture was incubated for 15 min at 37 °C, and kept on ice before use. The filter binding assay was performed as described below[33]. EF-Tu was twofold serially diluted in GTP charging buffer to final concentrations ranging from 0.094 to 12 µM, and Cy5-Gly-$tRNA^{Gly}$ was used at a concentration of 12 nM. The mixture was incubated for 10 min at RT in a 12.5 µl volume, 1.25 µl of 0.1 mg/ml RNase I was added, and incubated for 1 min at RT. Immediately, 0.625 µl of 4 mg/ml unfractionated yeast tRNA (Sigma) was added as quencher and coprecipitant, followed by 17.5 µl of 10% (W/V) ice-cold trichloroacetic acid (TCA). The precipitate was collected on a nitrocellulose filter presoaked in 5% TCA by vacuum and washed with ~1 ml of 5% ice-cold TCA. The filter was washed in 95% ethanol at room temperature for 5 min and imaged for Cy5 fluorescence on a Typhoon 9410 Variable Mode Imager (GE Healthcare Life Sciences) with ImageQuant software (Molecular Dynamics).

Electrophoretic mobility shift assays were performed as previously described[31]. Briefly, tRNAs were incubated in 8 mM GTP at concentrations ranging between 0.4 to 1 µM in 50 mM Tris-acetate, pH 6.0, 5 mM BME, 70 mM $NH_4Cl$, 7 mM $MgCl_2$, and 30 mM KCl. Then, the tRNAs were mixed with GTP charged EF-Tu at concentrations ranging between 50 to 200 nM and 0 to 8 µM, respectively, in 10 µl final volume. The reaction was loaded onto a 10% polyacrylamide mini-gel containing 10 mM MES (2-(N-morpholino)ethanesulfonic acid), pH 6.0, 65 mM ammonium acetate, 10 mM magnesium acetate, 10 µM GTP, and electrophoresed in the same buffer at a power of <4 W at 4 °C for 1.5 to 4 h.

**Single-molecule FRET**. Single-molecule FRET were performed using a prism-based TIRF (total internal reflection fluorescence) microscope[22,47,50,51]. All smFRET movies were acquired at 100 ms time resolution using an intensified CCD camera (I-Pentamax, Princeton Instruments)[50,51]. Quartz slides with a microfluidic channel containing inlet and outlet ports for buffer exchange were assembled as in previous work[50]. In short, the surface of the microfluidic channel was coated with biotinylated-bovine serum albumin followed by streptavidin for immobilizing biotinylated T-box RNA molecules. We flowed 20–50 pM of fluorophore-labeled T-box RNA molecules in standard buffer into the channel for surface immobilization. Any unbound molecules were washed out with the same buffer. Doubly labeled T-box riboswitch molecules or T-box riboswitch:tRNA complexes were excited with 532 nm green laser for measuring FRET and later with 640 nm red laser to check for the presence of Cy5 (acceptor) in order to distinguish low/zero FRET molecules from the ones missing an acceptor fluorophore. An enzymatic oxygen scavenging system consisting of 5 mM protocatechuic acid (PCA) and 50 nM protocatechuate-3,4-dioxygenase (PCD) along with 2 mM Trolox was included in the smFRET buffer to extend the lifetime of fluorophores and to prevent photoblinking of the dyes[52]. The raw movies were processed using IDL (Research Systems) to extract smFRET time traces and later analyzed using custom-written Matlab (The Math Works) scripts. Individual smFRET time traces displaying single-step photobleaching, a signal-to-noise ratio of >4:1, a total (donor + acceptor) fluorescence intensity of >300 (arbitrary units), and a total fluorescence duration of >10 s were manually selected. These selection parameters ensure that only single molecules and not aggregates or background impurities are analyzed. The ratio $I_A/(I_A + I_D)$, where $I_A$ and $I_D$ represent the background-corrected fluorescence intensities of the acceptor (Cy5) and donor (Cy3) fluorophores, respectively, was used to calculated the FRET value at every time point in smFRET traces. The first 50 frames from all the smFRET traces in each condition were used to plot ensemble FRET ratio histograms using OriginLab 8.1 and fit with a sum of Gaussian functions to estimate their mean FRET values and relative populations. Distances (R) between the fluorophores were then calculated from Eq. (1) below using the mean FRET (E) values and an $R_0$ value of 54 Å for Cy3–Cy5 dye pair[22]:

$$E = \frac{1}{1 + (R/R_0)^6}. \tag{1}$$

A minimum of 100 molecules were observed in all experiments to ensure a statistically significant measurement where the results had essentially converged.

**Single-molecule tRNA binding kinetic assays**. Single-molecule binding assays were performed similarly as above using prism-based TIRF microscopy by directly exciting both Cy3 and Cy5 with green and red lasers (direct excitation mode) or by exciting Cy3 only (FRET mode) while recording both Cy3 and Cy5 fluorescence. For the kinetic binding assays using direct excitation with the red laser, Cy5-labeled $tRNA^{Gly}$ concentrations ranging from 12.5 to 25 nM were used, whereas higher concentrations (up to 50 nM) were used for detecting tRNA binding to Stem-I using FRET. For experiments involving the complex of Cy5-Gly-$tRNA^{Gly}$ with EF-Tu (GTP charged), the two molecules were initially mixed in standard buffer and incubated at room temperature for 10 min, followed by the addition of PCA, PCD, and Trolox of the oxygen scavenger system, to yield final concentrations of 12.5 nM Cy5-Gly-$tRNA^{Gly}$ and 2 µM EF-Tu. Due to a higher background, traces with Cy5 S/N ratio of >3:1 where the binding events were clearly discernible were manually selected for analysis. The same oxygen scavenging system described above was used to minimize the photobleaching rate. Multiple movies with observation times of 10–15 min were acquired at 100 ms time resolution, during which there was

minimal focal drift as evident from a similar Cy5 intensity in the binding events. Kinetic movies using charged tRNA were acquired within a time window of 45 min to 1 h to minimize deacylation of the tRNA, which is expected to be slow under the slightly acidic pH of our buffer. Single-molecule time traces showing Cy3 intensity and with at least two binding events were only taken for further analysis. Of note, a minority of T-box molecules (~40%) never appeared to bind tRNA and were deemed inactive/misfolded. Traces showing binding were idealized with a two-state (bound and unbound) model using segmental $k$-means algorithm in QuB[53,54]. From the idealized traces, dwell times of tRNA in the bound ($t_{bound}$) and unbound ($t_{unbound}$) states were obtained. Cumulative bound and unbound dwell-time distributions were plotted and fitted with single-exponential or double-exponential functions to obtain lifetimes in the bound ($\tau_{bound}$) and unbound ($\tau_{unbound}$) states, respectively. The tRNA dissociation rate constant ($k_{off}$) was calculated as the inverse of $\tau_{bound}$, whereas the association rate constant was calculated by dividing the inverse of $\tau_{unbound}$ by the concentration of free Cy5-tRNA$^{Gly}$ or Cy5-Gly-tRNA$^{Gly}$ used.

For measurement of $k_{off}$ for the ultra-stable complexes, T-box:tRNA complexes were formed by mixing 100 nM T-box riboswitch with 500 nM Cy5-tRNA$^{Gly}$ in standard buffer and incubated on ice for 10 min. For experiments involving the complex of EF-Tu and Gly-tRNA$^{Gly}$, these two molecules were incubated at concentrations of 3.2 μM and 230 nM, respectively, for 10 min at room temperature in standard buffer. The complex was then incubated at a concentration of 150 nM tRNA with 75 nM riboswitch on ice for 30 min. Riboswitch complexes of 50–100 pM in standard buffer containing OSS was flowed onto the slide to achieve good surface density, followed by incubation for 1 min. Unbound molecules were washed away by flowing 200 μl of standard buffer with OSS. The molecules were imaged using both green and red lasers at low laser power using 200 ms exposure and images were acquired at 20 s intervals over 20 min. The number of spots in the Cy5 channel was identified and their decrease over time was fit with a single-exponential function to estimate the tRNA bound lifetime of the ultra-stable complexes.

**Data availability**. Data supporting the findings of this study are available within the article (and its Supplementary information files) and from the corresponding authors upon reasonable request.

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

## Acknowledgements

We thank members of the Walter laboratory for helpful discussions, and Dr. Julia Widom for providing a Matlab script for estimation of the slow $k_{off}$. E.P.N. and N.G.W. acknowledge support from NIH grant R01 GM115857.

## Author contribution

K.C.S., J.C.-V., N.G.W., and E.P.N. designed the experiments; M.M. and E.P.N. made the T-box riboswitch and tRNA constructs; K.C.S. and J.C.-V. performed the single-molecule experiments and analyzed the data; C.M. performed and analyzed some of the Gly-tRNA$^{Gly}$:EF-Tu:GTP affinity measurements; K.C.S., N.G.W., and E.P.N., wrote the manuscript; K.C.S., J.C.-V., N.G.W., and E.P.N. interpreted the data and reviewed the text and figures. All authors read and approved the manuscript.

## Additional information

Competing interestThe authors declare no competing interests.

