## [Peer Review File · Nature Communications]

Reviewers' comments:

Reviewer #1 (Remarks to the Author):

The manuscript by Walter and coworkers analyzes the important regulatory process of T-box riboswitching in gram-positive bacteria. Many genes that are associated with aminoacyl-tRNA synthetases are regulated via binding of the uncharged tRNA to the untranslated leader sequence of the corresponding mRNA. Crystal structures revealed two of the crucial tRNA mRNA recognition features that are direct bp and stacking interactions of the tRNA anticodon and its elbow to the preorganized and rather rigid mRNA fold domain. However, the structural details for the recognition of the uncharged CCA end with the antiterminator/terminator region that is crucial for the switching mechanism has remain elusive. For the glycyl T-box riboswitch, Walter and coworker now succeeded to reveal the mechanism of this recognition by dissecting the dynamics of the complex binding process. They revealed that reversible docking of the tRNA to the mRNA occurs via elbow and anticodon. This process is dynamic and fast (in the order of seconds). Only when the tRNA is uncharged the tRNA becomes trapped by stable tRNA anchoring onto the downstream antiterminator. This governs a hierarchical sensing mechanism that uses dynamic global binding of one tRNA side followed by readout of its aminoacylation status “by snap-lock based trapping”, to use the authors’ pictorial word creation.

This biophysical study is certainly worth to be considered for publication. The smFRET is beautiful and technically sound. The reference and control measurements are meaningful (i.e. different construct lengths, EF-Tu bound tRNA, etc.). The whole story provides a complete picture. I very much like the high quality of the Figures including Figure 6 which culminates in an excellent summary of the kinetics of the system and makes is memorable.

Minors:

- The non-covalent labeling works (surprisingly) very well. Nevertheless the number of total molecules (~100) is small compared to other smFRET studies. The authors may wish to comment on that.
- The authors used glycyl charged tRNA and mention that it does not hydrolyze rapidly. This is also a bit surprising but Fig. 5 speaks for itself. Can the authors distinguish between 2' and 3' charged tRNA (amino acid migration can/will occur); and if yes, do they observe a difference in kinetics (is 2'-charged tRNA accepted by the riboswitch)?

Reviewer #2 (Remarks to the Author):

The manuscript of Suddala et al. describes single-molecule fluorescence experiments to probe how T-box riboswitches discriminate aminoacyl from deacylated tRNAs to regulate translation through conformational modulation of RNA. While the experiments presented are well-performed and generally well analyzed, the presentation here is convoluted and confusing. Perhaps the push to present this in a compact format is detrimental to the different experimental schemes, and final model presented by the authors. The figures, in particular, are difficult to follow. The authors shift labeling schemes, and their schematics are very difficult to follow in terms of experiment design. For each experimental setup, they should have a clear schematic of where labels are located, and what signals we are observing. Their current schematics are too colored and I really could not sometimes see where the labels were located. In summary, I am not at all supportive of publication of the manuscript in Nature Communications. In fact, I believe this paper format does a disservice to the author's work; a more specialized journal would be a better target.

Here are some more detailed points that hopefully will help the authors improve their presentation:

Broadly speaking, the paper is not written in a manner that is accessible to a wide audience, as would be appropriate for Nature Communications. For example, the authors do not mention what gene(s) are controlled by *B. subtilis* glyQS, nor is the exact role of this particular element in vivo described in the text. The abstract is especially heavy in detail, most notably the 3rd, 4th and 5th sentences.

Starting in Figure 1, there is a clear discrepancy between the level of detail shown for the termination-opposing and termination-promoting conformations of the T-box riboswitch. A secondary structure diagram (and an appropriately nuanced discussion) is provided for the antitermination conformation, but only a single cartoon is shown for the termination-promoting state in panel a. This cartoon leads the reader to surmise that an intrinsic terminator is generated by folding of elements directly 3' of the T-box, however, none of these points are explained well in the text. The authors should explain what is known about the termination-promoting conformation to provide readers with the context required to understand their data.

Regarding the fits for $K_{off,fast}$, $K_{off,med}$ and $K_{off,slow}$ shown in Figures 2-4. It would be nice, however, to also see fits to single and double exponential models (where appropriate) to understand why as many as three states are necessary to fully explain the data. The authors also mention the existence of equilibrium measurements, however no direct comparisons are made to these data (e.g. is the ~ 300 nM K_d measured for tRNA/Stem I binding comparable to observations by others?).

In Figure 5, the gel shifts in Figure 5f are far from convincing and almost look like overloaded bands – more explanation would be helpful here. Alternatively, the authors could conduct RNase protection assays +/- EftU with both unmodified Gly-tRNA^{Gly} and Cy5-tRNA^{Gly} (necessary for understanding whether the modification impacts binding), move this data to the main text, and shift Figures 5e-f to the supplement. The experiment in Figure 5i, conducted with uncharged tRNAs and EF-Tu, would be a nice control.

In the discussion, the authors mention in vitro kinetic studies of pausing sites - a search of Net-seq data on *B. subtilis* pause sites (Larsen et al., Science 2014) may also be informative.

Reviewer #3 (Remarks to the Author):

THIS HAS BEEN UPLOADED AS FILE NCOMMS-17-28777.DOCX

NCOMMS-17-28777-T Hierarchical mechanism of amino acid sensing by the T-box riboswitch
Suddala et al

T-box riboswitch bind t-RNA to regulate the synthesis of aminoacyl-t-RNA synthetases. The T-box stem I binds to t-RNA with good affinity of around 150 nM. Two structures of the complex have been solved by crystallography in the Ferré D'Amare and Ke labs, showing two main contacts between the L-shaped riboswitch and the t-RNA. These occur between the T/D loop region of the t-RNA with the double T-loop of the riboswitch, and basepairing between the anticodon of the t-RNA and a complementary sequence in the riboswitch.

In the present study the authors have done a number of single molecule fluorescence experiments to characterize the dynamics of the interaction between various riboswitch constructs and t-RNA^{Gly} either charged or not with glycine. Essentially all the experiments reported are measurements of association and dissociation rates calculated from dwell times, and values of FRET efficiency. From these the authors conclude that there is an initial binding with fast k_{on} and k_{off} rates, and a more long-lived state that can occur with the un-aminoacylated t-RNA. This fixes the anti-terminator so allowing the initiation of transcription to occur. It is proposed that contact between the antiterminator region of the T-box RNA and the 3'-terminus of the t-RNA is the key interaction

leading to this slow koff rates, and this should be affected by the presence of the covalently-attached amino acid.

This is new information, but I am unsure whether this represents sufficient advance to merit publication in a high profile journal.

A major concern about this study has to be the nature of the t-RNA. The fluorescent RNA was made by ligation of commercially synthesized RNA, and therefore I presume lacking the many covalent modifications of natural RNA. I am concerned that this would prejudice the dynamics of the system.

A second concern is the conversion of FRET efficiency to distance throughout the paper, and rather prominently in many of the figures. Energy transfer results from a vector interaction between the two transition dipoles and therefore depends of the orientation as well as distance. In other words, the Förster factor R_0 cannot be assumed to be a constant as the authors assume. It is reasonable to look at changes in FRET efficiency with time, but all the conversions to distance should be taken out; they are just not reliable, especially since a lot of the values are actually quite similar.

Minor comments, and clarity issues.

At times I struggled to follow this manuscript. For example, the authors repeatedly talk about the "core" of the riboswitch (first introduced on page 5), yet this is never defined or shown in a figure. This probably seems obvious to the labs working daily on this project, but less so to the general reader. A key part of the argument is experiments with various truncated forms of the riboswitch. This would be so much easier to follow if the experimental figures (2-5) had small schematics of the construct used in each.

It was not clear to me what was the role of the K-turn in the bistable structure. This is at least 20 Å from the t-RNA, and certainly not part of the contact.

Fig. 5 e and f. With the Cy5 attached to the t-RNA how can the authors measure anything from the electrophoresis, which just makes a smear in the gel ? This is rather unconvincing. And on a point of clarity again, on the figure the only difference between parts e and f is color. Could Cy5 not be written onto part f somehow to save the reader having to burrow into the legend to figure it out ?

p11 the authors write : "consistent with an encoded actuation of the bistable K-turn motif embedded at the base". What is an "encoded actuation" please ?

I guess its not my job to tidy up the English here, but the manuscript does contain rather a lot of split infinitives, which rather grate.

Reviewer #1:

The manuscript by Walter and coworkers analyzes the important regulatory process of T-box riboswitching in gram-positive bacteria. Many genes that are associated with aminoacyl-tRNA synthetases are regulated via binding of the uncharged tRNA to the untranslated leader sequence of the corresponding mRNA. Crystal structures revealed two of the crucial tRNA mRNA recognition features that are direct bp and stacking interactions of the tRNA anticodon and its elbow to the preorganized and rather rigid mRNA fold domain. However, the structural details for the recognition of the uncharged CCA end with the antiterminator/terminator region that is crucial for the switching mechanism has remain elusive. For the glycyl T-box riboswitch, Walter and coworker now succeeded to reveal the mechanism of this recognition by dissecting the dynamics of the complex binding process. They revealed that reversible docking of the tRNA to the mRNA occurs via elbow and anticodon. This process is dynamic and fast (in the order of seconds). Only when the tRNA is uncharged the tRNA becomes trapped by stable tRNA anchoring onto the downstream antiterminator. This governs a hierarchical sensing mechanism that uses dynamic global binding of one tRNA side followed by readout of its aminoacylation status “by snap-lock based trapping”, to use the authors’ pictorial word creation.

This biophysical study is certainly worth to be considered for publication. The smFRET is beautiful and technically sound. The reference and control measurements are meaningful (i.e. different construct lengths, EF-Tu bound tRNA, etc.). The whole story provides a complete picture. I very much like the high quality of the Figures including Figure 6 which culminates in an excellent summary of the kinetics of the system and makes is memorable.

We are glad that the reviewer found our work interesting and appreciate their recommendation that the work be published. We also thank the reviewer for the kind words on the quality of the summary figure (now Figure 8).

Minors:

- The non-covalent labeling works (surprisingly) very well. Nevertheless the number of total molecules (~100) is small compared to other smFRET studies. The authors may wish to comment on that.

Non-covalent labeling using DNA oligonucleotide hybridization works well, given the high thermodynamic stability (calculated $T_m \sim 59^\circ\text{C}$) of the 14-nt long DNA-RNA hybrid we used. For smFRET measurement between the two ends of the T-box core using LNA-Cy5 and DNA-Cy3 oligonucleotides, we have more than 200 molecules. However, for our smFRET studies using the Cy5-labeled tRNA, its dynamic interaction with the T-box riboswitch with short bound times (<35 s) leads to tRNA dissociation from most of the T-box riboswitch molecules. This precludes observation of FRET between the tRNA and the ends of the T-box riboswitch since most of the tRNA dissociates before we can begin imaging (5 min after immobilization of the complexes onto the slide). Although more molecules can lead to smaller uncertainties in the measured values, our experience in smFRET indicates that 100 molecules is a decent number that reports on the equilibrium distribution of the system. We find that having more molecules only leads to subtle changes in the FRET histograms; we now note this convergence in the Methods section on page 18.

- The authors used glycine charged tRNA and mention that it does not hydrolyze rapidly. This is also a bit surprising but Fig. 5 speaks for itself.

Glycylated tRNA is fairly stable compared to many other charged tRNAs (please see ref #28). A recent study (please see ref #29 by Zhang et. al) showed that it is stable for at least 30 min at pH 7.0. Our experiments were performed at a slightly acidic pH of 6.1, at which the stability of the aminoacylated tRNA is expected to be further enhanced. In addition, when measuring binding kinetics using charged-tRNA, we limited our imaging time to ~1 h to minimize any hydrolysis of the

ester bond so that our imaging is done under homogeneous conditions. These details are now noted on page 19.

Can the authors distinguish between 2' and 3' charged tRNA (amino acid migration can/will occur); and if yes, do they observe a difference in kinetics (is 2'-charged tRNA accepted by the riboswitch)?

Bacterial Gly-AARs belong to the type/class II family of aminoacyl tRNA synthetases that attach amino-acid on the 3'-end only. Since we used E. coli S100 extracts for charging tRNA, most of it should be 3' charged tRNA. However, we cannot exclude small amounts (a few percent) of 2' charged tRNA to arise after intramolecular migration and, unfortunately, cannot distinguish it, a challenge shared by the field.

Reviewer #2:

The manuscript of Suddala et al. describes single-molecule fluorescence experiments to probe how T-box riboswitches discriminate aminoacyl from deacylated tRNAs to regulate translation through conformational modulation of RNA. While the experiments presented are well-performed and generally well analyzed, the presentation here is convoluted and confusing. Perhaps the push to present this in a compact format is detrimental to the different experimental schemes, and final model presented by the authors.

We thank the reviewer on the nice comments on our experiments and their analysis. However, we are puzzled by the descriptors “convoluted and confusing”, which is exactly the opposite of the impression of Reviewer #1. We have further attempted to present the material systematically with sufficient detail to allow for critical analysis while including additional details in the supporting information so as not to distract the reader. We understand that the number of riboswitch constructs and complexes is not small, but we have endeavored to reduce the potential for complexity in the text presentation through the use of diagrams/schematics in the figures as much as possible. We have further simplified the figures by dividing them into smaller ones and also included schematics in all figures.

The figures, in particular, are difficult to follow. The authors shift labeling schemes, and their schematics are very difficult to follow in terms of experiment design. For each experimental setup, they should have a clear schematic of where labels are located, and what signals we are observing.

The different labeling schemes are used to measure FRET between different regions of the T-box:tRNA complex. We have now divided the larger figures (previous Figs. 3 and 5) into smaller ones for better clarity. We also have added schematics to Figures 6 and 7 (previously Fig. 5) and included more detailed descriptions in the figure legends.

Their current schematics are too colored and I really could not sometimes see where the labels were located.

Again, this impression seems to be a matter of taste. The different colors in the schematics are used to highlight the various key motifs of the T-box riboswitch. Nevertheless, we have now changed/reduced the orange color of the schematics to dark grey for better contrast and easier viewing of the various motifs in the T-box.

In summary, I am not at all supportive of publication of the manuscript in Nature Communications. In fact, I believe this paper format does a disservice to the author's work; a more specialized journal would be a better target.

We respectfully disagree with the reviewer's opinion. The reviewer's comment on suitability appears to be rooted in the aesthetics of the manuscript and not the quality or significance of the work. Our study employs single-molecule FRET and novel single-molecule binding assays on the interaction of two large and structured RNAs, making it relevant not only to researchers working on riboswitches, but also those in many other scientific areas, including single-molecule biophysics, non-coding RNAs, RNA biophysics, transcription, and macromolecular binding. Therefore, we strongly believe that our work is well-suited for publication in a journal with broad readership like Nature Communications.

Here are some more detailed points that hopefully will help the authors improve their presentation: Broadly speaking, the paper is not written in a manner that is accessible to a wide audience, as would be appropriate for Nature Communications. For example, the authors do not mention what gene(s) are controlled by *B. subtilis glyQS*, nor is the exact role of this particular element in vivo described in the text.

The glyQS Tbox riboswitch controls expression of the glyQS gene that codes for the enzyme glycine tRNA synthetase. We have added this sentence to the introduction on page 2:

“The archetypal glyQS T-box riboswitch from *B. subtilis* controls the expression of glyQS gene encoding glycyl-tRNA synthetase in response to the ratio of charged to uncharged tRNA^{Gly} in the cell¹³.”

We have also added a few more sentences at the very beginning of the introduction about riboswitches and their general mechanism of action for novice readers, all highlighted in red.

The abstract is especially heavy in detail, most notably the 3rd, 4th and 5th sentences.

Naturally the Abstract contains an information-heavy summary of any study. To avoid confusion, we have clarified our findings and avoided the use of complex jargon in the Abstract so that it can be understood by general readers. This is also true for sentences 3-5 that, as the reviewer mentions, contain important scientific detail. The only technical terms of ‘tRNA’, ‘Stem-I’ and ‘T-box’ are all proper descriptors that are elaborated in the Introduction.

Starting in Figure 1, there is a clear discrepancy between the level of detail shown for the termination-opposing and termination-promoting conformations of the T-box riboswitch. A secondary structure diagram (and an appropriately nuanced discussion) is provided for the antitermination conformation, but only a single cartoon is shown for the termination-promoting state in panel a. This cartoon leads the reader to surmise that an intrinsic terminator is generated by folding of elements directly 3' of the T-box, however, none of these points are explained well in the text. The authors should explain what is known about the termination-promoting conformation to provide readers with the context required to understand their data.

We agree with the reviewer that some details are missing about the terminator sequence and secondary structure. We have now included these details in Fig. 1 and expanded our discussion in the Introduction. Furthermore, we have cited several recent reviews by experts in the field for more details on the T-box riboswitch (in particular references #3, 4, 17).

Regarding the fits for Koff,fast, Koff,med and Koff,slow shown in Figures 2-4. It would be nice, however, to also see fits to single and double exponential models (where appropriate) to understand why as many as three states are necessary to fully explain the data.

For Stem-I, the bound dwell times are all short and uniform so that the k_{off} is obtained by a good fit to a single-exponential corresponding to a short bound life-time of ~4.5 s. In contrast, for the core riboswitch a second longer, ~35 s, dwell-time appears that is due to the interaction of the tRNA 3'-end with the antiterminator. The kinetics here are clearly bi-exponential. This is also true for the Δ DTM variant. The $k_{off,slow}$, however, is measured from a different kinetic assay to assess this extremely slow dissociation rate; due to the separation of time scales, this separate dataset can be fit to single-exponential to obtain the very long lifetime. As requested, we have now added comparative single- and double-exponential fits to the dwell time distributions of k_{off} , where applicable (see Fig. 4c, 5f and Supplementary Fig. 5). We thank the reviewer for this helpful suggestion.

The authors also mention the existence of equilibrium measurements, however no direct comparisons are made to these data (e.g. is the ~300 nM K_d measured for tRNA/Stem I binding comparable to observations by others?).

As per this suggestion, we have added a comparison of the K_d measurements obtained in our study with those measured previously, see page 5:

*“This K_d is within ~2-3 fold of the values previously measured with isothermal titration calorimetry (ITC) for the *B. subtilis* and *O. theyensis* glyQS Stems-I under different buffer conditions^{15,25}.”*

In Figure 5, the gel shifts in Figure 5f are far from convincing and almost look like overloaded bands – more explanation would be helpful here.

Reviewer 2 (and also reviewer 3) indicated they were not overwhelmed by the quality of the electrophoresis data in Figure 5 showing the interaction of EF-Tu with the Cy5-tRNA. We now have further clarified this result and interpretation. In the particular experiment shown in Figure 5f, there is a broad band that the reviewers describe as a "smear" or "overload". This band is only visible in the presence of EF-Tu, and shows a sharp increase between 1 and 2 μ M EF-Tu. These findings indicate that the K_d is between those concentrations. This K_d range is further supported by quantification of the free and unbound forms of Cy5-Gly-tRNA at 1 μ M, as depicted in former Supplementary Figure 5g. We now provide an updated and cleaner version of this figure instead in the main text (see revised Fig. 7b), to show that the broad band that the reviewers viewed as a smear or overload is due to a shift, caused by EF-Tu, of Gly-tRNA and not the uncharged tRNA. We also now show that this shift does not occur with BSA.

Alternatively, the authors could conduct RNase protection assays +/- EFTU with both unmodified Gly-tRNAGly and Cy5-tRNAGly (necessary for understanding whether the modification impacts binding), move this data to the main text, and shift Figures 5e-f to the supplement. The experiment in Figure 5i, conducted with uncharged tRNAs and EF-Tu, would be a nice control.

The purposes of both the gel shift and RNase protection assays were to a) show that EF-Tu forms a complex with Cy5-Gly-tRNA; and b) estimate a lower limit of the concentration of EF-Tu needed to have most of this tRNA in complex with the protein. We have now included RNase protection controls, as new Fig. 7c and 7d, that clearly indicate that EF-Tu protects the charged Cy5-Gly-tRNA^{Gly} but not the uncharged Cy5-tRNA^{Gly}. In addition, our updated Fig. 7b gel image shows the shift due to EF-Tu more clearly, which is not seen with uncharged-tRNA or BSA negative control. Together, the revised gel shift and RNase protection assays strongly support that EF-Tu does bind to charged Cy5-labeled Gly-tRNA^{Gly}. Once the K_d range was estimated for the Cy5 Gly-tRNA, we performed our single molecule experiments accordingly to achieve near-full complexation with EF-Tu. Notably, comparison of this equilibrium with the K_d of EF-Tu and unlabeled Gly-tRNA would not yield additional information useful to interpret or design the fluorescence study. As per the

reviewer's suggestion, we now have included the RNase protection assay in the main text Fig. 7 (previously Fig. 5), promoting undergraduate student Collin Marshall from the Acknowledgements to co-authorship.

In the discussion, the authors mention in vitro kinetic studies of pausing sites - a search of Net-seq data on *B. subtilis* pause sites (Larsen et al., Science 2014) may also be informative.

We greatly appreciate the author pointing out the Net-seq data on B. subtilis pause sites from the Science paper. As per the suggestion, we have scrutinized these data and included the result as a new Supplementary Figure 6 and discuss it on page 13 as follows:

“However, under high intracellular NTP concentrations and in the presence of cellular transcription-modulating factors this long pause is absent, as evidenced by the high-throughput NET-seq data from Larson et al.⁴¹, while a relatively long pause is still detected at residue 178 (Supplementary Fig. 6). This pause would position the T-box sequence just outside the polymerase exit channel (~15 residues upstream). We propose that the pause at this location will allow time for the T-box sequence to interrogate the aminoacylation status of the bound tRNA, relying on the base pairing interaction with the tRNA 3'-end. The base pairing would then promote folding of the antiterminator once the pause ends, before the rest of the terminator sequence is made. Assuming an average rate of transcription by bacterial RNAP of ~20-25 nt/s^{20,42}, in the absence of any RNAP pausing before position 178, only a short (~4 s) time window would be available for Stem-I to capture tRNA and stabilize the weak antiterminator before the RNAP reaches this key decision point.”

Reviewer #3:

T-box riboswitch bind t-RNA to regulate the synthesis of aminoacyl-t-RNA synthetases. The T-box stem I binds to t-RNA with good affinity of around 150 nM. Two structures of the complex have been solved by crystallography in the Ferré D'Amare and Ke labs, showing two main contacts between the L-shaped riboswitch and the t-RNA. These occur between the T/D loop region of the t-RNA with the double T-loop of the riboswitch, and basepairing between the anticodon of the t-RNA and a complementary sequence in the riboswitch.

In the present study the authors have done a number of single molecule fluorescence experiments to characterize the dynamics of the interaction between various riboswitch constructs and t-RNAs, either charged or not with glycine. Essentially all the experiments reported are measurements of association and dissociation rates calculated from dwell times, and values of FRET efficiency. From these the authors conclude that there is an initial binding with fast k_{on} and k_{off} rates, and a more long-lived state that can occur with the un-aminoacylated t-RNA. This fixes the anti-terminator so allowing the initiation of transcription to occur. It is proposed that contact between the antiterminator region of the T-box RNA and the 3'-terminus of the t-RNA is the key interaction leading to this slow k_{off} rates, and this should be affected by the presence of the covalently-attached amino acid.

This is new information, but I am unsure whether this represents sufficient advance to merit publication in a high profile journal.

We would like to underscore the high relevance our new insights and novel single molecule fluorescence assays provide. The T-box riboswitch system is a remarkable, RNA-based gene regulatory system exemplifying complex interactions between two large and structured non-coding RNAs, representing a model for many such trans-RNA interactions. Its uniqueness and importance is also evident from a slew of recent publication on the T-box riboswitch in eminent journals (see, e.g., references #14-16, 29, 34). Most of these previous studies showed the biochemical requirements for efficient functioning of the system and the recent partial crystal structures highlighted how tRNA is recognized by Stem-I. However, since the riboswitch functions during transcription, the kinetics of its tRNA interaction are critical for understanding its gene regulatory

mechanism. Our work here expands our understanding of the system significantly beyond all recent work by, for the first time, revealing the temporal dynamics of the tRNA interaction with the T-box, while also discovering the functional consequences of the tRNA being aminoacylated with a small amino acid. Although the mechanism of the T-box riboswitch is dependent on dynamic binding of the tRNA through three distinct contacts, these transient interactions have not previously been probed and dissected. Our work here not only shows a stable solution conformation of the system exquisitely dependent on the aminoacylation state of the tRNA, but also uncovered fast and complex multi-exponential binding kinetics uniquely accessible by single molecule probing. These fast kinetics show how the T-box architecture evolved to quickly capture a tRNA and ultra-stabilize it only in the absence of aminoacylation, utilizing the downstream antiterminator domain.

A major concern about this study has to be the nature of the tRNA. The fluorescent RNA was made by ligation of commercially synthesized RNA, and therefore I presume lacking the many covalent modifications of natural RNA. I am concerned that this would prejudice the dynamics of the system.

All structural studies and most biochemical studies of the T-box riboswitches so far have been performed using similarly in vitro transcribed tRNA lacking any modifications. These studies showed robust tRNA dependent antitermination and also high affinity T-box:tRNA interactions (please see references #3, 4, 18 for details). tRNA modifications determined to have the greatest impact in translation are present in the anticodon stem-loop. tRNA^{Gly,GCC} has no natural modifications in the anticodon stem-loop so that in vitro transcribed tRNA^{Gly} should in fact reflect the natural tRNA with respect to the ASL as a major binding site of the T-box. However, enhancements to free tRNA folding or stability that modifications in the T(psi)C loop may confer would not be accounted for in the in vitro synthesized tRNA and may possibly have some subtle effects on the dynamics; this, however, will not affect any of our conclusions, which are directly compatible with the plethora of prior studies of the system in the absence of tRNA modification.

A second concern is the conversion of FRET efficiency to distance throughout the paper, and rather prominently in many of the figures. Energy transfer results from a vector interaction between the two transition dipoles and therefore depends of the orientation as well as distance. In other words, the Förster factor R_0 cannot be assumed to be a constant as the authors assume. It is reasonable to look at changes in FRET efficiency with time, but all the conversions to distance should be taken out; they are just not reliable, especially since a lot of the values are actually quite similar.

The distances shown in the figures are rough estimates ($\pm 5 \text{ \AA}$) and, therefore, we consistently refer to them as ‘approximate’ (using the \sim symbol) distances when discussing them in the paper. We have converted the FRET values to distances to give a sense of the physical separation between different regions and the spatial dimensions of the T-box riboswitch:tRNA complex. The Förster radius R_0 is a constant for a donor-acceptor pair for a given system under specific buffer conditions. The value varies slightly depending on the local microenvironment of the dyes, which are dependent on the labeling positions and proximal RNA sequences. For the well-characterized Cy3-Cy5 dye pair, the γ value, which accounts for the differences in the quantum yield and detection efficiencies of the dyes and needs to be measured to obtain accurate distances, is close to 1. It is therefore common practice in smFRET studies to talk about ‘approximate’ distances obtained from the Cy3-Cy5 pair, as we have done here.

The fact that the distances appear similar is due to the large size of the complex and not just an artifact of the FRET to distance conversions. We also observe that the labeled positions do adopt close distances ($\sim 36 \text{ \AA}$) transiently (see Fig. 3b, top trace), representing a relative change as the reviewer suggests. In contrast, the ~ 0 FRET values we observe for certain complexes represent distances $> 80 \text{ \AA}$ due to the diminishing change in FRET outside an $\sim 30\text{--}70 \text{ \AA}$ window, so we do not

*show them in the figures. Therefore, it is not true that all our estimated distance values are quite similar. In fact, on page 6, for the Stem-I to antiterminator FRET measurement, we see three quite distinct distances (53 Å, 79 Å and 36 Å). Nevertheless, as per the suggestion we have removed the distances from the schematics, but do mention them in the results as approximate distances, to give readers a sense of the dimensions of the complex in solution that can be compared with distances emerging with more structural studies. We also now explicitly state on page 6 that these are “corresponding to Stem-I-to-antiterminator distance **estimates** of ~53 Å, ~79 Å and ~36 Å (each ±5 Å)”.*

Minor comments, and clarity issues.

At times I struggled to follow this manuscript. For example, the authors repeatedly talk about the "core" of the riboswitch (first introduced on page 5), yet this is never defined or shown in a figure. This probably seems obvious to the labs working daily on this project, but less so to the general reader.

As per the reviewer’s suggestion, we have now defined the “Core” riboswitch early in the manuscript in the Introduction on page 3 as “including Stem I, III and the antiterminator stem-loop“, as well as in the legend for Figure 1. We thank the reviewer for their suggestion.

A key part of the argument is experiments with various truncated forms of the riboswitch. This would be so much easier to follow if the experimental figures (2-5) had small schematics of the construct used in each.

Figures 2-4 now have the schematics for each variant of the riboswitch. In addition, we now ensure that the figure title and the legend clearly state the variant used in those experiments and we included a label in the schematics. Following the reviewer’s suggestion, we have also included schematics for former Figure 5 (split now into Figures 6 and 7) and thank the reviewer for their suggestion.

It was not clear to me what was the role of the K-turn in the bistable structure. This is at least 20 Å from the t-RNA, and certainly not part of the contact.

The K-turn is below the specifier loop region and not part of the contact, as visible in Fig. 1 and in the schematics. When measuring FRET between the base of Stem-I and the bound tRNA, the kinked and extended conformations result in ~0.4 and ~0 FRET states. The extended conformation of the K-turn moves the Cy3-labeled DNA oligonucleotide more than 40 Å away from the bound tRNA, which results in a ~0 FRET state. The function of the bistable K-turn motif and the assignment of the 0 and 0.4 FRET states are stated in the Discussion on page 11 as follows:

*“Upon kinking the RNA backbone by ~120°, the tRNA acceptor stem will be positioned closer to the antiterminator, as necessary for amino acid sensing (Fig. 8a). Comparison of the corresponding distances (~64 Å between the base of Stem-I and bound tRNA variable loop) in the kinked conformer of the K-turn from the crystal structure¹⁵ with the extended conformation observed using NMR spectroscopy (with a corresponding between the distance of >90 Å) in a proximal portion of an apo *tyrS* T-box Stem-I²⁶ supports assignment of the observed ~0.4 and ~0 FRET states to the kinked and extended conformations, respectively.”*

Fig. 5 e and f. With the Cy5 attached to the t-RNA how can the authors measure anything from the electrophoresis, which just makes a smear in the gel ? This is rather unconvincing.

We appreciate this concern and refer to our response to reviewer #2’s similar comment above.

And on a point of clarity again, on the figure the only difference between parts e and f is color. Could Cy5 not be written onto part f somehow to save the reader having to burrow into the legend to figure it out ?

As suggested, we now have included 'Cy5' in the gel description for Fig. 5 panel f (updated as Fig. 7b).

p11 the authors write : "consistent with an encoded actuation of the bistable K-turn motif embedded at the base". What is an "encoded actuation" please?

I guess its not my job to tidy up the English here, but the manuscript does contain rather a lot of split infinitives, which rather grate.

"encoded actuation" means "genetically encoded functioning/kinking" (of the bistable K-turn motif). We have simplified this sentence by removing "encoded", as well as a few other sentences for ease of understanding and thank the reviewer for their suggestion.

REVIEWERS' COMMENTS:

Reviewer #1 (Remarks to the Author):

The authors have addressed all my concerns very carefully and I am also fully satisfied with the changes made by the authors in response to the other referees.